# New Insights on the Uptake and Trafficking of Coenzyme Q

**DOI:** 10.3390/antiox12071391

**Published:** 2023-07-06

**Authors:** Michael D. Guile, Akash Jain, Kyle A. Anderson, Catherine F. Clarke

**Affiliations:** Department of Chemistry & Biochemistry and the Molecular Biology Institute, University of California, Los Angeles, CA 90059, USA; mdguile@g.ucla.edu (M.D.G.); akashj@g.ucla.edu (A.J.); kyleanderson24@g.ucla.edu (K.A.A.)

**Keywords:** coenzyme Q, lipid trafficking, membrane contact sites, mitochondria transport, ubiquinone

## Abstract

Coenzyme Q (CoQ) is an essential lipid with many cellular functions, such as electron transport for cellular respiration, antioxidant protection, redox homeostasis, and ferroptosis suppression. Deficiencies in CoQ due to aging, genetic disease, or medication can be ameliorated by high-dose supplementation. As such, an understanding of the uptake and transport of CoQ may inform methods of clinical use and identify how to better treat deficiency. Here, we review what is known about the cellular uptake and intracellular distribution of CoQ from yeast, mammalian cell culture, and rodent models, as well as its absorption at the organism level. We discuss the use of these model organisms to probe the mechanisms of uptake and distribution. The literature indicates that CoQ uptake and distribution are multifaceted processes likely to have redundancies in its transport, utilizing the endomembrane system and newly identified proteins that function as lipid transporters. Impairment of the trafficking of either endogenous or exogenous CoQ exerts profound effects on metabolism and stress response. This review also highlights significant gaps in our knowledge of how CoQ is distributed within the cell and suggests future directions of research to better understand this process.

## 1. Introduction

Coenzyme Q (CoQ), or ubiquinone, is a vital lipid needed for generation of energy and a crucial antioxidant that preserves membrane structure and function. Although it is synthesized within the mitochondria of eukaryotes, it is present in all cellular membranes and is also a component of lipoproteins. CoQ_10_ (the isoform synthesized by humans and most mammals) is an enormously popular supplement and a medically important therapeutic. Yet, its insolubility in water means it must be trafficked between membranes via lipid vesicles, micelles, chaperones, or transport proteins. The pathways and mechanisms that move this lipid from one membrane to another are still largely undefined. Elucidating the pathways of this trafficking will aid our understanding of the functional roles of CoQ and may lead to improved therapeutics.

In this introductory section, we review the structure and function of CoQ, discuss the genes necessary for its biosynthesis, and provide a short overview of CoQ deficiencies and therapeutic strategies. In the sections that follow, we aim to summarize the work in several model organisms used to investigate how CoQ is trafficked. We will discuss how the yeast *Saccharomyces cerevisiae* has been used to identify proteins required for the uptake and trafficking of CoQ_6_ (the isoform of CoQ synthesized by *S. cerevisiae*), both to and from mitochondria. Mammalian cell culture models have also been used to identify pathways involved in the assimilation of exogenous CoQ_10_. We will then address how studies in mice, rats, and guinea pigs have been used to identify the uptake and assimilation of exogenous dietary CoQ isoforms.

Human studies and clinical trials addressing the bioavailability and pharmacokinetics of CoQ_10_ supplements and formulations are the topics of many extensive reviews, and are not discussed here. For readers interested in these topics we recommend the following reviews: [1,2,3,4,5,6,7,8,9]. Additionally, we will not cover the biosynthesis of CoQ in eukaryotic organisms such as *Schizosaccharomyces pombe*, *Arabidopsis thaliana*, *Caenorhabditis elegans*, and in microbes such as *Escherichia coli* as there are excellent reviews that cover these topics [10,11,12].

### 1.1. Structure and Function of CoQ_n_

CoQ is a redox-active lipid composed of a fully substituted quinone ring and polyisoprenyl tail of variable length that is species dependent (CoQ*_n_*where *n* is the number of isoprene units). Humans and most mammals synthesize CoQ_10_, mice and rats produce CoQ_9_ with minor amounts of CoQ_10_, *C. elegans* CoQ_9_, *E. coli* CoQ_8_, and *S. cerevisiae* CoQ_6_ [12,13,14,15]. CoQ can be found in the oxidized state, the semi-reduced state (ubisemiquinone, CoQH·), or the fully reduced state (ubiquinol, CoQH_2_). CoQ isoforms with polyisoprene tails six units or longer are generally considered to reside at the midplane of the bilayer [16]; however, there are conflicting interpretations regarding the orientation of CoQ isoforms in lipid bilayers [17]. Less hydrophobic isoforms such as CoQ_2_ and CoQ_4_ readily diffuse across non-contiguous membranes. Meanwhile longer isoforms, including CoQ_6_, CoQ_9_, and CoQ_10_, are unable to move between lipid vesicles in the absence of a transporter [18,19].

Mitochondrial CoQ plays an essential role in aerobic respiration and can be reduced by oxidoreductases involved in several metabolic pathways (Figure 1a). At the inner mitochondrial membrane, CoQ shuttles electrons along the electron transport chain by accepting electrons from NADH:ubiquinone oxidoreductase (Complex I) and succinate dehydrogenase (Complex II), and donating them to the cytochrome *bc*_1_ complex (Complex III). Cytochrome *c* then transports electrons from Complex III to cytochrome oxidase (Complex IV), which directly reduces O_2_ into water. The free energy from this electron transfer is used to transport protons from the matrix side to the intermembrane space, ultimately creating a proton-motive force used by ATP synthase to generate ATP [20,21,22]. Mitochondrial CoQ is also reduced by several other dehydrogenases and thus participates in diverse metabolic pathways, including sulfide:quinone oxidoreductase (SQR) for catabolism of sulfide, dihydroorotate dehydrogenase (DHODH) for pyrimidine biosynthesis, choline dehydrogenase (CHDH) for choline oxidation, glycerol-3-phosphate dehydrogenase (GPDH) for the glycerol-3-phosphate shuttle, proline dehydrogenase (PRODH) for catabolism of proline, and electron-transferring-flavoprotein dehydrogenase (ETFDH) for oxidation of fatty acids, branched-chain amino acids, sarcosine, and dimethylglycine (Figure 1a) [23].

In mitochondrial and other biological membranes, CoQ functions as a powerful antioxidant and plays a role in cell maintenance. CoQ is the only endogenously synthesized lipid-soluble antioxidant and acts to inhibit lipid peroxidation from damaging cellular membranes, proteins, and DNA. CoQH_2_ acts as a chain terminator to inhibit both the initiation and propagation steps of autoxidation of polyunsaturated fatty acids (PUFAs) [24,25]. Moreover, CoQH_2_ is also capable of regenerating α-tocopherol (vitamin E) from the tocopheroxyl radical (Figure 1b) [26]. Recent studies indicate that the oxidized form of CoQ can also slow peroxyl radical addition reactions responsible for cellular damage incurred by electrophilic stress resulting from conjugated PUFAs, such as conjugated linoleic and conjugated linolenic acids [27].

CoQ at the plasma membrane is part of the plasma membrane redox system (PMRS), which combats oxidative stress, maintains cell bioenergetics when mitochondrial activity decreases, and is involved in cell growth and resistance to apoptosis (Figure 1c) [28,29]. The PMRS comprises NAD(P)H:quinone oxidoreductase 1 (NQO1), NADH-cytochrome *b*_5_ reductase (cyb5Red), CoQ, and other antioxidants, such as vitamin E and ascorbate. The PMRS functions to regulate cellular redox homeostasis through the maintenance of cytosolic NAD(P)^+^/NAD(P)H and CoQ/CoQH_2_ pools within the plasma membrane. CoQH_2_ generated by the PMRS can function as a non-competitive inhibitor of neutral sphingomyelinase, a membrane protein that releases ceramide from membrane sphingomyelin to activate ceramide-dependent caspases and elicit apoptosis in serum-deprived cells [29,30,31]. Additionally, ferroptosis suppressor protein 1 (FSP1) was recently identified as a CoQ reductase that regenerates CoQH_2_ at the plasma membrane to inhibit ferroptosis in a pathway that is distinct from the anti-ferroptotic glutathione peroxidase 4 (GPX4) system (Figure 1d) [32,33]. Intriguingly, vitamin K is also reduced by FSP1 and inhibits ferroptosis [34], while tetrahydrobiopterin may function as a co-antioxidant with CoQ to inhibit ferroptosis [35]

### 1.2. CoQ Biosynthesis, Mutations in COQ Genes, and Therapeutic Strategies

CoQ is synthesized at the inner mitochondrial membrane on the matrix side, but is found throughout all cellular membranes [36]. CoQ biosynthesis requires a high molecular mass complex, termed the CoQ synthome in yeast and complex Q in mammals, situated on the matrix side of the inner mitochondrial membrane (Figure 2) [37,38,39]. CoQ biosynthesis begins with formation of the aromatic ring precursor 4-hydroxybenzoic acid (4-HB). In yeast, 4-HB is derived from the shikimate pathway or from tyrosine [38]. Human cells, however, may derive 4-HB from phenylalanine or tyrosine due to the presence of phenylalanine hydroxylase [37]. Deamination of tyrosine produces 4-hydroxyphenylpyruvate (4-HPP). In humans, mouse, rat, and guinea pig, this step is thought to be catalyzed by tyrosine aminotransferase (TAT) or by mitochondrial alpha-aminoadipate aminotransferase (AATAD) [37]. In yeast there is considerable redundancy in the pathway of 4-HB synthesis; five aminotransferase enzymes can perform this deamination step [40]. In humans, hydroxyphenylpyruvate dioxygenase-like protein (HPLD) converts 4-HPP to 4-hydroxymandelate (4-HMA) [41]. 4-HMA is also a likely intermediate in the yeast pathway to 4-HB [40,42]. The additional steps producing 4-hydroxybenzaldehyde (4-Hbz) have yet to be determined in either yeast or human cells. In yeast, Hfd1 is required to oxidize 4-Hbz to 4-HB, and the human aldehyde dehydrogenase 3A1 (ALDH3A1) has also been shown to perform this step [37]. COQ2, or Coq2 in yeast, first attaches the variable length polyisoprenyl tail to 4-HB. The polyprenylated intermediate is then introduced to the CoQ synthome (or complex Q), where the COQ3-COQ9 polypeptides act in concert to synthesize CoQ (Figure 2) [10,38]. The final CoQ product is then transported to its intracellular destination in the mitochondria, plasma membrane, or other organellar membranes via poorly characterized pathways [37].

Mutations in the CoQ biosynthetic genes may result in a condition termed primary CoQ_10_ deficiency. Primary CoQ_10_ deficiency is a rare disease often associated with multisystem disorders [43]. However, deleterious mutations that produce a poorly functioning or non-functional polypeptide, can also result in isolated phenotypic consequences, affecting the kidney or central nervous system. Genetic or biochemical testing is required for comprehensive diagnosis [43,44]. Secondary CoQ_10_ deficiency may arise when mutations in genes not directly associated with CoQ_10_ biosynthesis affect cellular CoQ_10_ content [45], in response to drugs (such as statins), or as a result of aging [39]. Currently, the only treatment for CoQ_10_ deficiency is high-dose exogenous supplementation of CoQ_10_. While CoQ_10_ displays poor bioavailability, in some patients presenting with primary CoQ_10_ deficiency supplementation produced dramatic improvements in their long-term clinical phenotypes [6,46]. To improve such clinical outcomes, it is important to understand both the gastrointestinal (GI) uptake pathway of orally supplemented CoQ_10_, as well as the cellular uptake and intracellular distribution.

**Figure 2 antioxidants-12-01391-f002:**
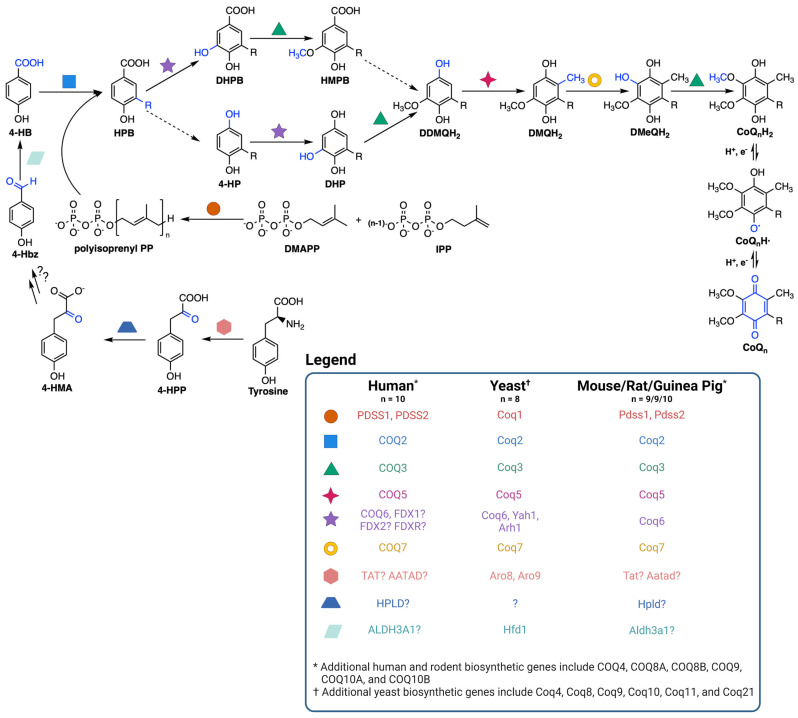
Coenzyme Q biosynthetic pathway. The dashed arrows indicate decarboxylation and hydroxylation steps that are catalyzed by unknown enzyme(s), thus resulting in an uncertainty in the order of reactions. Intermediates in the pathway include 4-HPP, 4-hydroxyphenylpyruvate; 4-HMA, 4-hydroxymandelate; 4-Hbz, 4-hydroxybenzaldehyde; 4-HB, 4-hydroxybenzoic acid; DMAPP, dimethylallyl pyrophosphate; IPP, isopentenyl pyrophosphate; HPB, 4-hydroxy-3-polyprenyl-benzoic acid; DHPB, 4,5-dihydroxy-3-polyprenylbenzoic acid; HMPB, 4-hydroxy-5-methoxy-3-polyprenylbenzoic acid; DHP, 4,5-dihydroxy-3-polyprenylphenol; DDMQH_2_, 2-methoxy-6-polyprenyl-1,4-benzohydroquinone; DMQH_2_, 2-methoxy-5-methyl-6-polyprenyl-1,4-benzohydroquinone; DMeQH_2_, 3-methyl-6-methoxy-2-polyprenyl-1,4,5-benzenetriol; *n* is the number of isoprene units in the polyisoprenyl tail. Yeast can also utilize *p*-aminobenzoic acid as a ring precursor, and its prenylation by Coq2 produces the early intermediate 4-amino-3-polyprenyl-benzoic acid (HAB, [38]). Adapted with permission from Wang et al. [44]. Created with Biorender.com (accessed on 28 June 2023).

The rescue of CoQ_10_-deficient animals supplemented with synthetic hydrophilic CoQ precursors or “bypass therapies” indicates that bioavailability is a key limitation of CoQ_10_ supplementation [6]. Bypass therapy for patients with primary CoQ deficiency is currently being studied as a potential alternative to supplementation with CoQ_10_ [47,48]. In bypass therapies, analogs of 4-HB are used as alternate ring precursors that function to “bypass” a blocked step in the CoQ biosynthetic pathway. For example, 2,4-dihydroxy-benzoic acid (2,4-diHB) partially restores CoQ biosynthesis in mice harboring mutations in *Coq7* (also termed *Mclk1*) [49]. Mice with certain mutations of Coq9 also respond to 2,4-diHB treatment [50]. However, 2,4-diHB also inhibits the endogenous CoQ biosynthetic pathway, possibly due to competition with natural substrates of the COQ enzymes upstream of the Coq7 hydroxylation step [49]. Bypass of human *COQ6* mutations using vanillic acid, a methoxylated analog of 4-HB, also has efficacy in a cultured cell model [51]. Due to the lack of a hydrophobic polyisoprenyl tail, the analogs of 4-HB are significantly more bioavailable than CoQ_10_, and the restoration of endogenous biosynthesis may achieve a more effective rescue [52].

## 2. The *S. cerevisiae* Model

*S. cerevisiae* (hereafter referred to as yeast) is a single-celled fungus that has been widely studied as a simple, eukaryotic model organism. Yeast cells are capable of both fermentation and aerobic respiration, and they divide rapidly through a budding process in which smaller daughter cells bud off from a larger mother cell. These cells are approximately five microns in diameter, between bacterial and human cells in size. Yeast cells can readily switch between haploid a and haploid α cells and mate to form diploid cells, which can further sporulate to produce four haploids [53]. Many eukaryotic cellular processes have been first identified and extensively characterized in yeast, including cell cycle regulation, vesicle trafficking, protein secretion, and autophagy. Many of these processes are highly conserved between yeast and humans. In fact, 47% of the 414 essential genes in *S. cerevisiae* with a 1:1 human ortholog can be functionally replaced by the corresponding human ortholog [54]. These well-characterized pathways, ease of genetic manipulation, and strong conservation between yeast and humans make yeast a powerful model organism.

Much work has been performed to elucidate the biosynthetic pathway and understand the cellular function of CoQ_6_ in yeast. The *COQ1–COQ11* genes encode mitochondrially targeted polypeptides that are required for efficient CoQ_6_ biosynthesis [37,38,39]. Strains with a deletion or deleterious mutation in one of the *COQ1–COQ9* genes, termed *coq* mutants, are unable to synthesize CoQ_6_ (Figure 3a). These *coq* mutants are thus incapable of cellular respiration and fail to grow in a non-fermentable medium. In most cases, expression of the human *COQ* ortholog in the respective yeast *coq* mutant rescues efficient CoQ_6_ biosynthesis and restores growth in a non-fermentable medium, highlighting the strong conservation of CoQ biosynthesis from yeast to humans. In addition to the presence of all the Coq polypeptides, efficient CoQ_6_ synthesis requires Coq3–Coq9 and Coq11 to form a mega-complex on the matrix side of the inner mitochondrial membrane, termed the CoQ synthome [38,39]. The formation of this complex is analogous to the Complex Q assembly that occurs in human cells [39]. The strong conservation of CoQ biosynthesis and many functions from yeast to humans suggest a conserved pathway of CoQ transport.

### 2.1. Growth Rescue of coq Mutants in a Non-Fermentable Medium Supplemented with Exogenous CoQ_6_

As previously stated, *coq* mutants are unable to synthesize CoQ_6_ and are thus respiratory deficient and fail to grow in a medium containing a non-fermentable carbon source (Figure 3a). Deletion of *COQ1* or *COQ2* that are involved in early steps of CoQ_6_ biosynthesis prevents the accumulation of early CoQ_6_ intermediates, 3-hexaprenyl-4-hydroxybenzoic acid (HHB) and 3-hexaprenyl-4-aminobenzoic acid (HAB), while deletion of any one of the *COQ3*-*COQ9* genes that are required for later biosynthetic steps accumulate these early CoQ intermediates (Figure 3b). Significantly, direct supplementation of exogenous CoQ_6_ restores respiration and growth of *coq* mutants [55,56], though the degree of rescue depends on the nature of the *coq* mutant. When supplemented with exogenous CoQ_6_ in non-fermentable media, *coq1Δ* and *coq2Δ* single mutants exhibit wild-type or near wild-type levels of growth, while each of the *coq3Δ–coq9Δ* single mutants exhibit growth that is significantly lower than wild-type cells (Figure 3c). Deletion of either *COQ1* or *COQ2* in the *coq3Δ* single mutant restores rescue to that of wild-type cells or to the *coq1Δ* and *coq2Δ* single mutants (Figure 3d), indicating that the accumulation of early CoQ_6_ intermediates, HHB and HAB, impairs growth rescue of *coq3Δ–coq9Δ* mutants [18]. It is possible that HHB/HAB act as detergents or may interfere with membrane trafficking steps needed for transit of exogenously added CoQ_6_. CoQ_6_ uptake is favored by the presence of peptone in the culture media [57]. The digested proteins in peptone likely bind CoQ_6_ and enable its entry into the aqueous compartment of endocytic vesicles. Following supplementation with exogenous CoQ_6_, the *coq* mutants also show a significant delay in growth compared to wild-type cells as they make the diauxic shift from fermentation to respiration [57]. Supplementation with either hydrophilic CoQ isoforms that can spontaneously diffuse across non-continuous membranes (CoQ_2_, CoQ_4_), or hydrophobic CoQ isoforms that cannot diffuse (CoQ_6_) are both capable of rescuing growth. Moreover, supplementation with CoQ_2_ can bypass defects in endocytosis and membrane trafficking [19,58]. The ability of yeast to uptake exogenous CoQ, transport it to the mitochondria, and in turn restore respiration in *coq* mutants makes yeast a powerful model organism to investigate CoQ uptake and transport.

Studies with supplementation of non-isoprenoid CoQ analogs indicate that the isoprenoid tail is essential for efficient uptake and growth rescue of *coq* mutants in a non-fermentable medium. Idebenone, decylQ, and MitoQ are three such analogs containing a ten-carbon alkyl tail (Figure 3e). The tail end of idebenone and MitoQ are further modified with a hydroxyl group or triphenylphosphonium moiety, respectively, with the latter exhibiting mitochondrial targeting. None of these three analogs were able to restore growth of *coq* mutants when supplemented in a non-fermentable medium [58]. Although decylQ and idebenone restore respiration in isolated mitochondria, it is speculated that mitochondrial accumulation of these analogs in whole cells is insufficient to support growth. Conversely, MitoQ accumulates in mitochondria of whole cells, but is poorly oxidized by Complex III and thus unable to restore respiration [58]. In fact, supplementation with high concentrations of decylQ inhibits growth rescue of *coq* mutants with CoQ_2_, perhaps through competition with an unidentified CoQ-binding protein [19]. Supplementation assays with analogs of CoQ_2_ where the prenyl tail has been modified show that each of the analogs tested are capable of restoring respiration in isolated mitochondria, but the presence of the double bond between C2 and C3 in the first isoprenoid unit of exogenously supplied CoQ is essential for efficient uptake in whole cells and growth in a non-fermentable medium [19]. Taken together, the decylQ impairment of growth rescue by CoQ_2_ and the structural requirement in the isoprenoid tail of exogenous CoQ suggest the presence of an essential protein that can distinguish subtle differences in CoQ structure for efficient delivery or retention of exogenous CoQ to mitochondrial respiratory complexes.

It is possible that the yeast Coq10 polypeptide may fulfill this role of a CoQ chaperone. Yeast Coq10 is peripherally associated with the matrix side of the inner mitochondrial membrane, but is not part of the endocytic pathway, the respiratory complexes, or the CoQ synthome [59]. The yeast Coq10 polypeptide contains a steroidogenic acute regulatory-related lipid transfer (START) domain, which binds CoQ_n_ isoforms of variable tail lengths and analogs of late-stage CoQ-intermediates such as DMQ_3_ [60]. Yeast Coq10 is needed for respiratory function, and may be involved in transport of newly synthesized CoQ_6_ to Complex III [61]. COQ10A and COQ10B are human orthologs of yeast Coq10, and partially rescue both the respiratory defect and sensitivity to lipid peroxidation of the *coq10Δ* yeast mutant [62]. Human STARD7 is structurally related to the yeast Coq10 and human COQ10 polypeptides, binds CoQ_4_ in vitro, and plays a role in vivo in transporting mitochondrial CoQ_10_ to the plasma membrane to suppress ferroptosis [63]. Human saposin B binds CoQ_10_, and is postulated to function as a transporter [64,65]. Mammalian CoQ transport proteins are discussed in Section 3.1.5. It is likely that many other CoQ transporter or chaperone proteins remain to be discovered.

### 2.2. Identification of Essential Genes in CoQ_6_ Uptake and Trafficking to Mitochondria Respiratory Complexes with the Use of Yeast ORFΔcoqΔ Double Mutants

#### 2.2.1. A Screen for Genes That Impact Growth Rescue by Exogenous CoQ

Previous studies performed by James et al. have sought to identify genes that influence the ability of exogenously supplemented CoQ_2_ and CoQ_4_ to support the respiratory growth of CoQ-less yeast [19]. Sixteen such genes were found by screening a library of ~4800 *ORFΔcoqΔ* double mutants. The deletion of these ORFs resulted in impaired growth rescue on a medium containing a non-fermentable carbon source when supplemented with exogenous CoQ_2_ or CoQ_4_. However, it is important to note that CoQ_2_ and CoQ_4_ have shorter isoprenoid tails than CoQ_6_, passively diffuse across noncontinuous membranes, and move independently of lipid trafficking mechanisms [18,19].

To further examine the role of these ORFs as well as other candidates in the uptake and trafficking of CoQ_6_, Fernández-Del-Río et al. tested whether exogenous CoQ_6_ restored growth in a screen of 40 *ORFΔcoq2Δ* double mutants in a non-fermentable medium [18]. This group of 40 genes included the 16 *ORFΔcoq2Δ* double mutants identified by James et al. [19]. The rescue assay compared the *ORFΔcoq2Δ* double mutant to the *coq2Δ* single mutant, and to the *ORFΔ* single mutant in a non-fermentable medium when supplemented with exogenous CoQ_6_ [18]. The degree of growth rescue in a medium supplemented with exogenous CoQ_6_ was shown to be dependent on the distinct *COQ* gene that was disrupted (see Section 2.1). Six *ORFΔcoq2Δ* double mutants demonstrated reduced growth in a non-fermentable medium with exogenous CoQ_6_ as compared to either the *coq2Δ* or the *ORFΔ* single mutants, suggesting a role for these ORFs in CoQ_6_ uptake and transport to the mitochondria. Fernández-Del-Río et al. reasoned that supplementation with the hydrophilic isoform CoQ_2_ should be able to circumvent the trafficking defect observed with CoQ_6_ in these mutants [18]. Indeed, the growth rescue afforded by CoQ_2_ supplementation in these *ORFΔcoq2Δ* double mutants far exceeded that of CoQ_6_. Such growth rescue assays of *ORFΔcoq2Δ* double mutants provide a powerful tool to identify genes that are required for the uptake of exogenous CoQ_6_ and its trafficking to mitochondria. However, more work is needed to unravel the role these gene products play in uptake and transport of exogenous CoQ_6_.

Interestingly, two genes, *RTS1* and *CDC10,* were identified as essential in both screens (Figure 4) [18,19]. Rts1 is a regulatory subunit of protein phosphatase 2A (PP2A), which plays a role in regulation of cell size control, the mitotic spindle orientation checkpoint, and septin ring organization and disassembly during cytokinesis [66,67]. It is possible that Rts1 in complex with PP2A plays a regulatory role in CoQ_6_ uptake and transport. *CDC10* is one of four core septins in yeast, a group of GTPases that assembles into filaments and higher order structures [68]. Septin filaments assemble into a septin ring at the bud neck in yeast and act as a scaffold to recruit other proteins to drive cytokinesis [69]. Septins have been shown to interact with a subset of endocytic proteins [70], and SEPT9, the human homolog of Cdc10, has been implicated in endosomal trafficking [71]. More work is needed to determine if the role of Cdc10 in CoQ_6_ uptake and transport is involved in endocytosis or another septin function.

The four other essential genes for uptake and trafficking of exogenous CoQ_6_ to respiratory complexes identified by Fernández-Del-Río et al. are *RVS161*, *RVS167*, *NAT3*, and *VPS1* (Figure 4) [18]. Rvs161 and Rvs167 are amphiphysin-like membrane proteins that form a heterologous dimer and promote membrane curvature [72]. Rvs161 and Rvs167 interact with several other proteins, including Vps1, at the plasma membrane for the vesicle scission step of endocytosis [73]. Vps1 is a GTPase involved in membrane fusion and fission. Vps1 also operates in other cellular processes, including vacuolar protein sorting, actin cytoskeleton organization, late Golgi retention of proteins, and peroxisome biogenesis [74]. Nat3 is a catalytic subunit of the NatB complex that mediates acetylation of the N-terminus of approximately 20% of proteins in yeast and humans [75,76]. The role that each of these genes plays in CoQ_6_ uptake and trafficking remains unclear.

Typically, mitochondria are excluded from schemes depicting endosomal trafficking [77]. This is curious because trafficking between the plasma membrane and mitochondria is reported to occur via clathrin-mediated endocytosis [78]. However, trafficking of proteins and lipids to mitochondria is likely mediated by extensive contact sites between mitochondria and other intracellular membranes (Figure 4) [79,80]. Fernández-Del-Rio et al. screened *ORFΔcoq2Δ* double mutants predicted to impair selected membrane contact sites such as the endoplasmic reticulum-mitochondria encounter structure (ERMES), the nucleus-vacuole junction (NVJ), and the vacuole–mitochondria patch (vCLAMP) [18].

However, none of these *ORFΔcoq2Δ* double mutants displayed impaired growth rescue when treated with exogenous CoQ_6_. These results could suggest that CoQ_6_ transport may occur independently of these membrane contact sites, or that transport through contact sites is redundant. For example, mitochondria dependence on phospholipid exchange has been shown to depend on either ERMES or vCLAMP [81]. Loss of ERMES stimulates vCLAMP and vice versa; however, loss of both is lethal [81]. Yeast genes that are redundant or essential for growth on a non-fermentable carbon source will not be detected by such exogenous CoQ_6_ growth-rescue screens. It is also possible that other lipid transport mechanisms operate independently of both endocytic and contact site-mediated trafficking.

#### 2.2.2. Rescue of *coq* Mutants with Exogenous CoQ Is Variable and Depends on the Genetic Background of the Yeast Strain

Growth rescue of *coq* mutants in a non-fermentable medium with exogenous CoQ_6_ is dependent on the yeast genetic background. For example, the commonly used BY4741/BY4742 laboratory yeast strain takes up and transports exogenous CoQ_6_ to the mitochondrial respiratory complexes, and was used in the screen to detect trafficking defects (Section 2.2.1) [18]. This is also the case for the W303-1B and CEN.PK2-1C yeast genetic backgrounds [82]. Yet, certain yeast strains harboring a *coq7Δ* null allele, such as the EG103*coq7Δ* or the FY250*coq7Δ* mutant yeast strains, fail to be rescued when the growth medium is supplemented with exogenous CoQ_6_. The content of CoQ_6_ in the gradient-purified mitochondrial fraction of these strains was only 35% and 8% of the content present in the mitochondria isolated from the EG103 and FY250 wild-type parental strains, respectively. However, the CoQ_6_ content in the plasma membrane fraction isolated from EG103*coq7Δ* and the FY250*coq7Δ* strains exceeded or equaled the content of CoQ_6_ normally present in the plasma membrane fraction of the parental strains [82]. In these mutant strains, the plasma membrane redox activity was augmented, suggesting that the exogenous CoQ_6_ present in the plasma membrane was functional [82]. Thus, it is likely that such strains unable to be rescued for growth by exogenous CoQ_6_ likely have a defect in trafficking the CoQ_6_ from the plasma membrane to the mitochondria [82]. Both the parental EG103 and the EG103*coq7Δ* mutant strains were shown to secrete carboxy-peptidase Y and accumulate small vacuoles, markers that characterize impaired membrane trafficking [57]. It would be important to test whether these mutants are rescued by supplementation with exogenous CoQ_2_, since this analog does not depend on lipid trafficking.

To test the idea that defects in endocytosis or endomembrane trafficking alter CoQ_6_ uptake and transport to the mitochondria, Padilla-López et al. interrogated *ORFΔcoq3Δ* double mutants whose *ORF* deletion, such as *ERG2*, *PEP12*, *TLG2*, or *VPS45*, disrupts several endocytic transport steps [57]. Supplementation with exogenous CoQ_6_ failed to rescue growth of these *ORFΔcoq3Δ* double mutants in a non-fermentable medium, suggesting that the endocytic and endomembrane trafficking pathways are involved in CoQ_6_ uptake and transport. Subcellular fractionation confirmed that each of these *ORFΔcoq3Δ* double mutants have decreased uptake of exogenous CoQ_6_ in mitochondria, Golgi, vacuole, and plasma membrane compared to WT and *coq3Δ* controls, further illustrating the impact of disrupting these trafficking pathways in CoQ_6_ uptake. Moreover, the low CoQ_6_ content in the plasma membrane of these mutants suggests that most of the exogenously supplied CoQ_6_ cannot be directly incorporated into the plasma membrane.

It must be noted that both *coq7Δ* and *coq3Δ* mutants accumulate HHB and HAB, early CoQ_6_ intermediates that act to impair growth rescue in a non-fermentable medium with exogenous CoQ_6_. Indeed, Fernández-Del-Río et al. showed that these *ORFΔcoq3Δ* double mutants had significantly less robust growth in a non-fermentable medium when compared to the *ORFΔcoq2Δ* double mutant counterpart (Figure 3c,d) [18]. More work must be performed in yeast to further determine the role of endocytosis and endomembrane trafficking in CoQ_6_ uptake and transport in yeast models. Previous work in the human promyelocytic HL-60 cell line has shown that both endogenously synthesized CoQ_10_ and exogenously supplied CoQ_9_ are distributed across membranes through the brefeldin A-sensitive endo-exocytic pathway (see Section 3.1.1) [83]; however, such experiments have not been performed in yeast.

### 2.3. CoQ_6_ Transport to and from the Mitochondria

It is likely that uptake and trafficking of exogenous and endogenous CoQ_6_ share common pathways. Recent studies suggest that the ERMES may play a role in both CoQ_6_ biosynthesis and distribution. ERMES is a tethering complex that forms at the membrane contact site between the ER and the outer mitochondrial membrane and is composed of two mitochondrial subunits embedded in the outer membrane (Mdm10 and Mdm34), an ER-localized subunit (Mmm1), and a cytosolic subunit (Mdm12) (Figure 4) [84]. The four subunits are functionally linked to phospholipid biosynthesis and calcium signaling [84]. Recent structural analyses show that ERMES forms bridge-like structures between the ER and mitochondrial outer membrane, consistent with providing a conduit for phospholipid transport [85]. The mitochondrial contact site and cristae organization system (MICOS), a mega complex that regulates cristae junction organization, has been shown to assemble near ERMES sites, and ERMES function is required for MICOS assembly [86]. The Mmm1-Mdm12 complex has been shown to mediate phospholipid transfer in vitro and mutations in *MMM1* and *MDM12* lead to impaired lipid transfer through ERMES in vivo [87].

Intriguingly, Coq polypeptides that are components of the CoQ synthome at the inner mitochondrial membrane are visualized as puncta or CoQ domains that lie adjacent to ERMES puncta at the outer mitochondrial membrane [88,89]. Disruption of ERMES decreases the number of CoQ domains [88,89]. Moreover, ERMES mutants show increased whole-cell accumulation of CoQ_6_ and CoQ_6_-intermediates but decreased content of CoQ_6_ in isolated mitochondria, suggesting that CoQ_6_ and CoQ_6_-intermediates accumulate in non-mitochondrial membranes [89]. Current observations suggest that destabilization of the CoQ synthome results in reduced sequestration of CoQ_6_ and CoQ_6_ intermediates in the mitochondria; alternatively, disruption of ERMES might lead to enhanced stability of non-mitochondrial CoQ_6_ [89]. Despite this role for ERMES in CoQ domain formation and CoQ_6_ production and distribution, direct transport of CoQ_6_ through ERMES has not yet been observed. Moreover, none of the four ERMES genes were identified as essential for CoQ uptake and transport in *ORFΔcoq2Δ* double mutants [18]. However, disruption of ERMES results in an expansion of vCLAMP contact sites, and deletion of both contacts has been shown to be lethal [81]. Thus, disruption of ERMES might increase compensatory contact sites to counteract potential defects in CoQ_6_ trafficking.

The idea that other membrane contact sites can compensate for lipid transport defects in ERMES mutants has been reinforced by recent structure/function studies of the VPS13 family of proteins. Yeast Vps13 is a 360 kDa protein composed of an extensive beta-sheet or “taco shell” lined by hydrophobic residues that can form a conduit for phospholipid transfer between organelles [90]. Yeast Vps13, Vps39, and Mcp1 form a mitochondria–vacuole contact site (Figure 4), and are required for the survival of yeast ERMES mutants [91]. Mcp1 is a mitochondrial outer membrane phospholipid scramblase, and partners with the Vps13 lipid bridge protein to re-equilibrate lipids as they are either inserted into or removed from the cytosolic leaflet of acceptor or donor membranes [92]. Vps39 is the vacuole fusion factor responsible for vCLAMP contact sites [81]. It is not yet known whether these bridge-like phospholipid transporters strategically positioned at contact sites also move neutral lipids, such as CoQ.

Recent work has identified yeast mitochondrial Cqd1 and Cqd2 proteins that influence cellular distribution of endogenous CoQ_6_ [93]. Intriguingly, both are inner membrane proteins that face the intermembrane space, and potentially provide a connection for transfer of CoQ_6_ from the inner membrane to the outer membrane and the ER (Figure 4). Disruption of *CQD1* or *CQD2* either diminishes or enhances the mitochondrial CoQ_6_ content, respectively, without altering overall cellular CoQ_6_ content [93]. Cqd1 and Cqd2 are members of the UbiB family of atypical kinases/ATPases. Coq8, an essential protein for CoQ_6_ biosynthesis and part of the CoQ synthome, is also a member of the UbiB family whose ATPase activity may be coupled to the extraction of hydrophobic CoQ_6_ intermediates from the inner membrane for subsequent processing by membrane-associated Coq polypeptides in the matrix [94]. Point mutations of both Cqd1 and Cqd2 at conserved protein-kinase-like (PKL) or UbiB motif residues disrupted their individual impact on CoQ_6_ distribution, indicating that Cqd1 and Cqd2 activities rely on atypical kinase/ATPase activity, though further biochemical work is needed to prove enzymatic activity. Furthermore, haploinsufficiency of the *CQD1* human ortholog *ADCK2* is reported to cause aberrant mitochondrial lipid oxidation and myopathy associated with CoQ_10_ deficiency [95].

Cqd1 participates in a contact site between outer and inner mitochondrial membranes (Figure 4), and overexpression of Cqd1 and Cqd2 elicits contact sites between the ER and mitochondria [96]. Taken in conjunction with the effect of Cqd1 on CoQ_6_ retention within mitochondria, and the effect of Cqd2 on promoting CoQ_6_ exit from mitochondria, these proteins likely play a role in CoQ_6_ trafficking to and from the mitochondria in yeast. It would be interesting to determine whether the Cqd1 complex is co-localized with the CoQ synthome and ERMES, as it provides the “missing link” between these two. Interestingly, deletion of both Cqd1 and Cqd2 in yeast restores the normal distribution of yeast CoQ_6_ [93], indicating that other trafficking mechanisms must operate. Hence, it is likely that other CoQ_6_ transporters contribute to trafficking.

Distribution of CoQ_6_ between mitochondrial and non-mitochondrial membranes plays important functional roles in mitochondrial respiration, response to oxidative stress, and redox control. The *cqd1Δ* single mutant has decreased mitochondrial CoQ_6_ content, grows slowly in a medium containing glycerol as the sole non-fermentable carbon source, has increased non-mitochondrial CoQ_6_ content, and shows enhanced resistance to treatment with PUFAs [93]. Conversely, the *cqd2Δ* single mutant has increased mitochondrial CoQ_6_ content, normal growth on a medium containing glycerol, and is more sensitive to treatment with PUFAs [93]. The trafficking of CoQ to non-mitochondrial membranes (including the plasma membrane) mediates resistance to ferroptosis, a type of cell death due to lipid peroxidation [32,33]. CoQ_6_ in the plasma membrane functions as a potent antioxidant and co-antioxidant (Figure 1b), and is also in the PMRS (Figure 1c). In respiratory deficient cells (yeast and HL-60 cultured cells), there is an increased content of CoQ at the plasma membrane [97,98]. This recruitment of CoQ to the PMRS enables cytosolic NADH to be oxidized to NAD^+^, together with export of the electrons to extracellular acceptors, such as ascorbate, and thereby maintains the ratio of cytosolic NAD^+^/NADH [99,100]. Hence, the pathways that control the trafficking of both endogenous and exogenous CoQ exert profound effects on metabolism and stress response.

## 3. Mammalian Cell Culture Models

Mammalian cell culture has been integral in the study of the uptake and distribution of CoQ for many decades. Among the reasons for using cell culture(s) to study CoQ uptake and trafficking are the high degree of specificity with which one can control their environment, the ease and accuracy of administration of nutrients, hormones, and in this case CoQ, the decreased physiological variability as compared to live animals. However, whether cells in culture accurately reflect their true physiological context is often difficult to determine.

In addition to uptake and trafficking of exogenous CoQ, cell culture has been used to study the CoQ synthome and endogenous CoQ biosynthesis [101]. The functional roles of CoQH_2_ as an antioxidant have also been characterized in culture [102,103], with a specific emphasis on plasma membrane-localized CoQ as a protectant against ferroptosis [32,33,104,105]. Cells in culture have likewise proven to be a powerful model for the study of CoQ uptake and distribution. In the following section, we discuss CoQ uptake and intracellular trafficking in epithelial, endothelial, blood, dermal, and HeLa cells. Subsequently, the supplementation and efficacy of CoQ_10_H_2_ and CoQ_10_ will be compared, before concluding with a discussion about cell receptors that mediate uptake of exogenous CoQ_10_.

### 3.1. Studies of CoQ Uptake in Mammalian Cell Culture

#### 3.1.1. Epithelial Cell Lines

Orally supplemented CoQ_10_ is first taken up by the epithelial cells of the digestive tract [2,9,106]. However, the mechanism by which enterocytes absorb CoQ_10_ is not well understood. Several reviews and studies state that the uptake of CoQ_10_ occurs by a simple process of passive diffusion [2,9]. A combination of passive diffusion and receptor-mediated uptake is responsible for absorption of fatty acids [107], vitamin E [108], and cholesterol [109] in enterocytes. Niemann–Pick C1-like 1 (NPC1L1) plays an important role in uptake of dietary cholesterol; its disruption in mice results in a 50% decreased cholesterol absorption [109]. Recent studies suggest that membrane protein transporters are also important for the uptake of CoQ_10_. The human intestinal epithelial cell line CaCo-2 provides a model for much of the recent research on CoQ absorption in the digestive tract. CaCo-2 is a colorectal adenocarcinoma cell line that is used to study transport across the intestinal epithelium, including the absorption of various hormones and small molecules such as insulin [110], anthocyanin [111], and calcitonin [112]. Among the advantages of this cell line is its formation of tight junctions, microvilli that form a brush border on the luminal or apical side, and transport proteins that allow the uptake and digestion or transport of nutrients to the basolateral side, all of which can simulate the in vivo digestion of molecules via oral supplementation (Figure 5) [113].

CaCo-2 cells can assimilate solubilized CoQ_10_ from digested oral supplements, such as CoQ_10_ in liposomes and micelles. Supplementation of CaCo-2 cells with CoQ_10_-infused liposomes altered the mRNA levels of 694 genes involved in signaling, metabolism, and general transport [106]. Many of these proteins are directly regulated by CoQ, such as Caspase-3, which is inhibited by CoQ_10_ at the plasma membrane [114]. CoQ_10_ uptake by CaCo-2 cells is enhanced when solubilized forms of CoQ_10_ were administered via simulated digestive micellization [115]. Eight different commercial CoQ_10_ products were mixed with fat-free yogurt and olive oil, and subjected to gastric conditions by treatment with hydrochloric acid and porcine pepsin. Following this treatment, lipase, bile extract, and porcine pancreatin were added to simulate digestion. This digestion micellized the CoQ_10_, which was then extracted from the aqueous layer of the resultant mixture. Of the eight CoQ_10_ products, four exhibited greater than 60% micellization in the aqueous layer: γ-Cyclodextrin complexed-CoQ_10_ tablet, liposomal CoQ_10_ solution, solubilized CoQ_10_ capsule, and a γ-Cyclodextrin/solubilized CoQ_10_ powder termed Hydro-Q-Sorb^®^ [115]. γ-Cyclodextrin is a water-soluble octasaccharide of particular interest to the optimization of CoQ uptake [116]. The sugar features eight glucose subunits interlinked to form a cylindrical shape, characterized by its hydrophilic exterior and hydrophobic interior, which can encapsulate CoQ_10_. γ-Cyclodextrin is one of many formulations of CoQ_10_ that have been assessed for their ability to enhance its bioavailability [117]. The liposomal CoQ_10_ and solubilized CoQ_10_ displayed the greatest rate of micellization, but it was Hydro-Q-Sorb^®^ that was absorbed most by CaCo-2 cells. Hydro-Q-Sorb^®^ exhibited a dramatic increase in CoQ_10_ content after a four-hour incubation when compared to incubation with the non-solubilized CoQ_10_ powder reference, despite Hydro-Q-Sorb^®^ containing only 20% of the concentration of CoQ_10_ as the reference powder [115]. The correlation between CoQ_10_ product micellization and CaCo-2 uptake was weak, suggesting that other factors regulate CoQ_10_ uptake besides micellar transport [115]. Regardless, there does exist a positive relation between the micellization of CoQ_10_ and its internalization in the CaCo-2 cell model, indicating the importance of lipid dispersion in CoQ uptake.

The CaCo-2 human intestinal epithelial cell model has been used to compare the relative effects of CoQ_10_ and CoQ_10_H_2_ uptake, as several studies postulate that CoQ_10_H_2_ is more bioavailable than CoQ_10_ [118]. Supplementation with CoQ_10_H_2_ poses a challenge because the hydroquinone undergoes rapid autoxidation under cell culturing conditions [119]. As such, studies designed to compare treatment with CoQ_10_H_2_ versus CoQ_10_ should monitor autoxidation of the CoQ_10_H_2_ during incubation conditions. Recently, Naguib et al. [120] found that CoQ_10_H_2_ complexed with γ-cyclodextrin remained stable in an aqueous salt solution for up to seven days, with the majority of the original CoQ_10_H_2_ not undergoing autoxidation to CoQ_10_, demonstrating that CoQ_10_H_2_ can be supplemented with high purity in this form. Failla et al. [118] investigated the relative uptake and transport of CoQ_10_ and CoQ_10_H_2_ across the CaCo-2 monolayer using a previously described simulated digestion method [115]. CoQ_10_ or CoQ_10_H_2_ was subjected to “digestion” and added to the apical medium. Following a four-hour incubation, the total cell-associated CoQ_10_ content and the total amount of CoQ_10_ transported into the basolateral medium were determined. Both apical uptake into CaCo-2 cells, and transepithelial transport was significantly higher for the micellized CoQ_10_H_2_ as compared to the micellized CoQ_10_ [118]. The authors showed that incorporation of CoQ_10_H_2_ into micelles during the digestion process was twice as efficient when compared with CoQ_10_ [118]. Non-complexed CoQ_10_ has been shown to have very low permeability across the CaCo-2 monolayer [118,121], indicating that in the absence of CoQ_10_ incorporation into micelles or liposomes, transport across the brush border is extremely inefficient. Thus, both physiological and pharmaceutical solubilization methods for CoQ_10_ (or CoQ_10_H_2_) are needed to enhance its absorption across the small intestine epithelium.

Small intestine epithelial cells express NPC1L1, a transport protein involved in cholesterol and vitamin E transport and metabolism (Figure 6a) [108,122,123]. Recent work has identified a role for NPC1L1 in the intestinal absorption of CoQ. NPC1L1 was overexpressed in Madin–Darby canine kidney (MDCK) epithelial cells to assess NPC1L1-mediated CoQ_9_ and CoQ_10_ transport [124]. MDCK cells are epithelial cells derived from kidney tissue of a cocker spaniel. NPC1L1 was expressed over 50-fold in human small intestine and duodenum compared to the kidney, making this mutant MDCK model reasonably analogous to that of intestinal CoQ uptake [125]. CoQ_10_ and CoQ_9_ uptake was found to increase significantly in NPC1L1-overexpressed MDCK cells, as compared to control MDCK cells. Furthermore, treatment with ezetimibe, a selective inhibitor of NPC1L1, decreased CoQ_10_ and CoQ_9_ uptake of NPC1L1-overexpressed cells back to control levels, implicating NPC1L1 as a CoQ-transport protein [124]. This study demonstrates the existence of intestine-specific proteins that function to increase CoQ_10_ absorption, among other lipids. However, the mechanism of NPC1L1-CoQ_10_ interaction remains unclear.

The HepG2 liver epithelial cell line has played an important role in the development of new strategies for exogenous CoQ_10_ uptake. Moschetti et al. [126,127] engineered a nanodisk composed of a phosphatidylcholine bilayer circumscribed by apolipoprotein-A1, where CoQ_10_ integrates readily within the midplane of the phosphatidylcholine bilayer. These nanodisk structures are defined as “reconstituted high-density lipoproteins” [128], and the CoQ_10_ is integrated into a disk-shaped bilayer. The solubilization efficiency of the nanodisk-CoQ_10_ was determined to be 97%, as compared to CoQ_10_ which is essentially insoluble when dispersed in the presence of just apo-A1 or phosphatidylcholine [126]. The authors then assessed the disk’s ability to transport CoQ_10_ into HepG2 cells. HepG2 cells exhibited a significant increase in mitochondrial CoQ_10_ content upon incubation with nanodisk-associated CoQ_10_ compared to empty nanodisks. The authors conclude this nanodisk to be a promising vehicle for exogenous CoQ_10_ uptake and distribution in liver epithelium. However, the study lacked a control where CoQ_10_ was supplemented in the absence of the nanodisk formation. This would have provided a useful point of comparison.

To further investigate the potential effectiveness of biomolecular nanodisks as a drug-delivery vehicle for CoQ_10_, cultured C2C12 myotubes were treated with supra-pharmacological doses of simvastatin, an HMG-CoA reductase inhibitor that decreases the production of polyprenyl diphosphate precursors, and nanodisk-CoQ_10_ [127]. Treatment with simvastatin caused a dose-dependent decrease in cell viability, reduced mitochondrial CoQ_10_ content, and reduced the oxygen consumption rate. However, co-treatment with simvastatin and nanodisk-CoQ_10_ increased mitochondrial CoQ_10_ content compared to simvastatin treatment alone and ameliorated the reduction in oxygen consumption rate. These results highlight the potential of nanodisks to dramatically improve the bioavailability of CoQ_10_. It would be interesting to assess the stability of CoQ_10_H_2_ in nanodisks and compare its uptake relative to that of the nanodisk-CoQ_10_. Animal studies are needed to determine the efficacy of nanodisks as a drug-delivery vehicle in vivo.

#### 3.1.2. Endothelial Cell Lines

Following absorption of CoQ_10_ into the bloodstream, endothelial cells are the next point of uptake. Endothelial cells form the monolayer that line the luminal surface of blood vessels and capillaries, mediating macromolecule transport to and from neighboring tissues. The endothelium is studied extensively in the uptake and distribution of drugs due to this essential role in post-digestive tract drug transport. Human umbilical vein endothelial cells (HUVECs), human corneal endothelial cells, and bovine aortic endothelial cells (BAECs) have been utilized as models for CoQ uptake.

HUVECs have been used to study cell response to CoQ_10_ as a protectant against oxidative stress. Treatment of HUVECs with Amyloid-β_1-42_ or with a sub-fragment of Aβ_1–42_ (Aβ_25–35_) causes oxidative stress and elicits cell death via apoptosis and necrosis [129]. Pre-treatment with exogenous CoQ_10_ (1–7 μM) was found to reduce reactive oxygen species (ROS) generation, inhibit the uptake of Aβ_25-35_, and prevent cell death [130]. The results suggest that the uptake and trafficking of exogenous CoQ_10_ is directed to sites essential in the quenching of free radicals within HUVECs [130].

Naguib et al. recently reported the uptake of CoQ_10_H_2_ in HCEC-B4G12 human corneal endothelial cells. HCEC-B4G12 cells exhibited a higher increase in total CoQ_10_ content when supplemented with CoQ_10_H_2_/γ-cyclodextrin as compared to non-complexed CoQ_10_ [120]. Furthermore, HCEC-B4G12 cells treated with ferroptosis-inducer erastin exhibited viability akin to that of untreated cells when incubated with as little as 1 µM CoQ_10_H_2_/γ-cyclodextrin, whereas HCEC-B4G12 cells treated with erastin and 100 µM non-complexed CoQ_10_H_2_ exhibited minimal rescue [120]. Uptake was also assessed by measuring ROS production in response to antimycin A, a mitochondrial electron transport chain Complex III inhibitor [120]. HCEC-B4G12 cells demonstrated a dose-dependent rescue of ROS generation in response to supplementation with CoQ_10_H_2_/γ-cyclodextrin, whereas non-complexed CoQ_10_H_2_ produced no significant change [120]. The authors report a promising level of CoQ_10_H_2_/γ-cyclodextrin uptake by this corneal endothelial cell line, pointing towards possible exogenous CoQ_10_H_2_ formulations involving γ-cyclodextrin to protect against oxidative stress-induced loss of function in human corneas.

BAECs incubated with non-complexed CoQ_10_ do not exhibit significant levels of uptake. When BAECs were treated with glucose oxidase, an enzyme that oxidizes glucose to produce hydrogen peroxide, incubation with MitoQ reduced the activity of apoptotic enzymes Caspase-3 and Caspase-9, indicating rescue from oxidative stress [131]. Conversely, incubation with CoQ_10_ did not reduce apoptotic activity, suggesting poor uptake of CoQ_10_ and/or impaired trafficking to sites essential for protection against oxidative stress [131]. Furthermore, when BAECs pre-treated with 1 µM Mito-Q were incubated in either glucose oxidase or hydroperoxides, complex I activity (as measured via the rate of NADH oxidation) exhibited significant rescue compared to the untreated control; whereas pre-treatment with 1 µM CoQ_10_ resulted in insignificant rescue in both glucose oxidase and hydroperoxide incubated BAECs [131]. Perhaps CoQ_10_H_2_/γ-cyclodextrin treatment could improve the uptake in BAECs, like that discussed for CaCo-2 and corneal epithelial cells.

#### 3.1.3. Human Blood Cells

The HL-60 cell line is a human myeloid leukemia cell derived from a patient with promyelocytic leukemia. These cells may be cultured in suspension. The trafficking of endogenously synthesized ^14^C-labeled CoQ_10_ was compared with the trafficking of exogenously added CoQ_9_ [83]. HL-60 cells were incubated with ^14^C-labeled 4-HB, the ring precursor of CoQ biosynthesis (Figure 2), and the content of ^14^C-labeled CoQ_10_ was determined in subcellular fractions over a period of 24 h. Fractions enriched in mitochondria contained more than 90% of the ^14^C-labeled CoQ_10_, as ascertained at time points from two to 24 h. Fractions enriched in ER and mitochondria-associated membranes (MAM) contained small amounts of ^14^C-labeled CoQ_10_ by two hours, and the plasma membrane fraction had detectable ^14^C-CoQ_10_ only after six hours [83]. HL-60 cells treated with exogenous CoQ_9_ showed an accumulation of CoQ_9_ in mitochondrial and MAM fractions, followed by its subsequent accumulation in ER and plasma membrane fractions. After twelve hours of incubation with CoQ_9_, the mitochondrial fraction contained the highest content of CoQ_9_, followed by the ER and MAM fractions with similar CoQ_9_ content, whereas the plasma membrane fraction exhibited significantly lower CoQ_9_ content [83]. The endo-lysosomal membrane fraction was most enriched in CoQ_9_, but this was not tracked during the time course. Treatment of HL-60 cells with brefeldin A, an inhibitor of vesicle formation and transport between the ER and Golgi, prior to the addition of ^14^C-4-HB resulted in a profound inhibition in the content of ^14^C-CoQ_10_ in all fractions, with the plasma membrane fraction being most decreased. It is interesting that inhibition of trafficking between the ER and Golgi had such a dramatic effect on the endogenous synthesis of CoQ_10_ within the mitochondria, and may indicate the importance of contact sites between the ER and mitochondria. Treatment of HL-60 cells with brefeldin A and CoQ_9_ resulted in most of the CoQ_9_ being sequestered in the endo-lysosomal fraction [83]. The high CoQ_9_ content in the endo-lysosomal fraction suggests the importance of the endo-lysosomal trafficking pathway in the uptake and assimilation of exogenous CoQ_9_. These findings indicate the importance of a functional endomembrane system in the synthesis of endogenous CoQ_10_ and in the transport of both exogenous CoQ_9_ and endogenous CoQ_10_.

A recent paper by Wani et al. [132] found exogenous CoQ_10_ (non-complexed) to ameliorate oxidative stress in erythrocytes and lymphocytes, suggesting its uptake. Erythrocytes and lymphocytes isolated from donor blood samples were exposed to TiO_2_ nanoparticles to induce oxidative stress. Upon treatment with TiO_2_, erythrocytes exhibited impaired ATPase activity and increased ROS production, and lymphocytes had severely depleted ATP levels and reduced mitochondrial membrane potential. Incubation with TiO_2_ in the presence of CoQ_10_ produced significant rescue of all markers of oxidative stress to wild-type or near-wild-type levels in both erythrocytes and lymphocytes [132]. These data indicate the ability of lymphocytes to take up and distribute exogenous CoQ_10_ to the mitochondria, where it restored mitochondrial membrane potential, ATPase activity, and ATP levels. Lymphocytes are also able to traffic exogenous CoQ_10_ to other cellular membranes, where it can slow lipid peroxidation induced by oxidative stress.

#### 3.1.4. Human Dermal Cells

Human dermal fibroblasts (HDFs) are specialized cells found in the dermis that play a vital role in the maintenance and repair of the skin. The use of HDFs is especially relevant to investigate the efficacy of CoQ_10_ as a commercially available skincare and anti-wrinkle product. HDFs exhibit significant levels of exogenous CoQ_10_ uptake, as measured by a reduction in ROS generation [133]. The authors used pollutant particulate matter to activate KU812 cells, a basophilic leukocyte cell line, and used the resulting conditioned medium as a stress treatment of HDFs. Such treatment elicits a proinflammatory response and ROS generation in HDFs. Following eight hours of treatment, the HDFs were then treated with 100 nM CoQ_10_. The exogenous CoQ_10_ affected a significant reduction in mitochondrial ROS generation [133]. These findings suggest the ability of HDFs to take up and traffic CoQ_10_ to the mitochondria.

HDFs have also been used to demonstrate an efficacious uptake of liposomal CoQ_10_. CoQ_10_ was suspended into liposomes via thin-film hydration, with a drug entrapment efficiency of 73.1% [134]. Administration of this liposomal-CoQ_10_ suspension to HDFs that were oxidatively stressed from hydrogen peroxide treatment resulted in a complete rescue of cell viability to the point of cell proliferation, whereas non-complexed CoQ_10_ administration of the same concentration caused no significant change in cell viability [134]. Similarly, quantification of ROS in these HDFs revealed that liposomal-CoQ_10_ supplementation significantly reduced ROS production, while supplementation with non-complexed CoQ_10_ produced no significant change [134]. The uptake of CoQ_10_ by HDFs is thus augmented by liposomal suspension, where the CoQ_10_ can restore antioxidant function.

HDFs exhibit improved uptake of exogenous CoQ_10_H_2_ versus CoQ_10_. Incubation of HDFs with simvastatin decreased CoQ_10_ levels by 29% [135]. A 24-h supplementation with 15 μg/mL CoQ_10_H_2_ caused a dose-dependent increase in total cellular CoQ_10_ content, up to an 87-fold increase compared to the vehicle control. In contrast, supplementation with 15 μg/mL CoQ_10_ resulted in just a 4.6-fold increase, a 95% reduction compared to CoQ_10_H_2_ [135]. Isolated mitochondria in HDFs treated with CoQ_10_H_2_ also exhibited a 160-fold increase in total CoQ_10_ content, compared to a 2.5-fold increase with CoQ_10_ treatment. Thus, treatment with CoQ_10_H_2_ resulted in enhanced mitochondrial CoQ_10_ content relative to treatment with CoQ_10_ [135]. The administration of CoQ_10_H_2_ to enhance uptake is promising, but the mechanisms that produce such high uptake relative to CoQ_10_ remain uncharacterized.

Keratinocytes are epidermal skin cells that constitute the majority of the human epidermis, essential in forming the primary defense against environmental damage. CoQ_10_ treatment has recently been shown to protect keratinocytes from oxidative stress-induced cell death, indicating its uptake [136]. HaCaT human keratinocytes derived from the foreskin were incubated with CoQ_10_ directly added to Dulbecco’s modified Eagle’s Medium (DMEM) and fetal bovine serum (FBS), prior to treatment with the free-radical generating compound 2,2′-azobis-(2-amidinopropane) dihydrochloride (AAPH). Interestingly, no rescue of cell viability was observed until the DMEM-CoQ_10_ medium was enhanced with CoQ_10_ premixed in FBS, which resulted in a significant rescue of cell viability in the keratinocytes [136]. Similarly, intracellular CoQ_10_H_2_ content increased significantly after incubation with the premixed CoQ_10_ medium, as compared to both the control and the original, insolubilized CoQ_10_ medium [136]. The authors suggest that uptake of CoQ_10_ by keratinocytes is augmented by the administration of CoQ_10_ in a premixed medium, where otherwise an inefficient uptake is observed.

Keratinocytes have proved useful in the characterization of a new iodine-labeled CoQ derivative that may allow monitoring of the biodistribution of exogenous CoQ [137]. The novel iodine-labeled CoQ_10_ (I_2_-Q_10_) consists of CoQ_10_ with two iodine atoms replacing the terminal carbon of the polyisoprenyl tail. Iodine fluoresces upon excitation via X-rays, allowing for quantification and localization of I_2_-Q_10_ with minimal overlap between other molecules. Human keratinocytes incubated in 50 µM I_2_-Q_10_ showed no signs of toxicity or cell death, with levels of CoQ_10_ uptake comparable to that of unlabeled CoQ_10_. X-ray fluorescence imaging of pelleted keratinocytes incubated in I_2_-Q_10_ revealed a cell-specific localization of the iodine fluorescence with no extra-pellet iodine leakage. Furthermore, incubation with 50 µM I_2_ resulted in a significant decrease in the pellet’s fluorescence, demonstrating poor uptake of elemental iodine and thus a mechanism for I_2_-Q_10_ uptake that is independent of iodine labeling uptake per se. The authors found exogenous I_2_-Q_10_ to distribute homogeneously over the keratinocyte cell, but co-localization with subcellular organelles, such as mitochondria, was not determined. Beyond the confirmation that CoQ_10_ is taken up by human keratinocytes, these data suggest a potential way to assess CoQ_10_ distribution using I_2_-Q_10_ in X-ray fluorescence imaging [137]. The technique was demonstrated to allow for quantitative analysis with no phenotypic changes due to the iodine labeling, and it shows promise in the investigation of mechanisms underlying exogenous CoQ uptake. It is also worth investigating whether the supplementation of with I_2_-Q_10_ restores respiratory function in a CoQ-deficient cell culture model, as some CoQ derivatives suffer from a loss of function upon modification of the tail (e.g., Mito-Q).

#### 3.1.5. HeLa Cells

HeLa cells, derived from cervical cancer cells, are the oldest and most generally used human cell line in medicine to date. Recent work in HeLa cells has identified STARD7 as a phosphatidyl transfer protein involved in the export of endogenous CoQ_10_ from the mitochondria to the plasma membrane, where it plays a role in ferroptosis suppression. STARD7 is localized to both the mitochondrial intermembrane space and the cytosol (Figure 6a). Deletion of STARD7 causes a nearly 5-fold increase in ferroptosis in response to treatment with PUFAs and erastin [63]. STARD7 is postulated to deliver mitochondrial CoQ_10_ to the plasma membrane, where it is reduced by FSP1 to CoQ10_10_H_2_ and suppresses ferroptosis. Expression of a cytosol-localized form of STARD7 in HeLa cells significantly decreased cell death in response to treatment with erastin and PUFAs. Inhibition of either CoQ_10_ biosynthesis or FSP1 function completely abolished resistance to ferroptosis, resulting in wildtype-levels of cell death in response to oxidative stress [63]. Expression of the mitochondria-localized form of STARD7 alone resulted in a significant decrease in plasma membrane-associated CoQ_10_, while the mitochondrial CoQ_10_ content was unaffected. The introduction of cytosol-localized STARD7 rescued plasma membrane-associated CoQ_10_ content, suggesting that STARD7 functions as a CoQ_10_ transporter in both the mitochondria and cytosol [63].

HeLa cells have also been used to study the effect of supplemented CoQ_10_ on sulfide oxidation. SQR is the initial enzyme in the sulfide oxidation pathway, whose impairment can lead to increased ROS. CoQ_10_ is an essential cofactor for SQR and protects the cell against oxidative stress. Silencing of *COQ8A* RNA in HeLa cells decreased the CoQ_10_ content by approximately 50% and led to enhanced ROS production. Supplementation of CoQ_10_ via Hydro-Q-Sorb decreased ROS production to near-control levels [138]. However, when SQR was knocked down in these CoQ_10_-deficient cells, exogenous CoQ_10_ no longer decreased ROS production [138]. This finding indicates that the supplementation of CoQ_10_ rescued ROS levels due to CoQ_10_-SQR activity, which requires CoQ_10_ localization to the inner mitochondrial membrane. Thus, HeLa cells could take up the Hydro-Q-Sorb CoQ_10_ and traffic it to the inner mitochondrial membrane, where it rescued sulfide oxidation.

HeLa cells have been utilized in the characterization of a lipopeptide-based surfactant, caspofungin (CF), that has been shown to solubilize CoQ. CF can form micellized nanoparticles with CoQ_10_ up to 200 nm in diameter, with a critical micelle concentration of 50 µM [139]. Wang and Hekimi recently found that CF-solubilized CoQ_10_ is a highly efficient formulation for exogenous supplementation in HeLa cells. Upon incubation for one hour, the CoQ_10_ content in human HeLa cells increased significantly when treated with CoQ_10_-CF as compared to CoQ_10_, demonstrating the complex’s rapid uptake. Recent results further demonstrated that treatment with CoQ_10_-CF restored maximal respiration in CoQ_10_-deficient human fibroblasts derived from a patient harboring a homozygous mutation in the *COQ7* gene [140]. The mechanism for CF micellization of CoQ_10_ remains unknown, and the potential for CoQ_10_-CF as a therapeutic to treat CoQ_10_ deficiencies in patients warrants investigation.

#### 3.1.6. Mouse Embryonic Fibroblasts (MEFs)

Wang et al. found that treatment with CoQ_10_-CF can increase the uptake of exogenous CoQ_10_ in MEFs in a dose-dependent manner. Incubation of MEFs for 72 h with 20 μM CF and 0.25 μM CoQ_10_ produced a near-doubling in basal and maximal oxygen consumption rate as compared mouse embryonic fibroblasts incubated in just 10 μM CF without exogenous CoQ_10_ [139]. After just a one-hour incubation with the CoQ_10_-CF complex, MEF CoQ_10_ content was approximately 3-fold higher when compared to MEFs incubated with free CoQ_10_ for 48 h, indicating the rapid uptake of the complex.

### 3.2. Cell Receptors for Exogenous CoQ Uptake and Transport

#### 3.2.1. Receptor-Mediated Uptake of CoQ

Receptor-mediated endocytosis plays a key role in the selective uptake of extracellular molecules, including CoQ_10_. Wainwright et al. [141], used an in vitro blood–brain barrier (BBB) endothelial cell model of CoQ_10_ deficiency, and identified lipoprotein-associated uptake and efflux mechanisms regulating CoQ_10_ distribution across the BBB (Figure 6b). Porcine brain endothelial cells (PBECs) were grown in monolayer on a permeable membrane (see Figure 5), and transport of CoQ_10_ was assessed from both the apical to basal (A to B, blood-to-brain side), and basal to apical (B to A, brain-to-blood side). The transport of non-complexed CoQ_10_ across porcine brain endothelial cells (PBEC) was extremely low. Pre-treatment of CoQ_10_ with bovine plasma-derived serum led to its association with the VLDL/LDL lipoprotein fractions and increased CoQ_10_ transport across the PBEC monolayer by 4-fold, suggesting that CoQ_10_ may be transported across the BBB as part of a lipoprotein complex. Transport of serum-treated CoQ_10_ in the A to B direction matched transport in the B to A direction, indicating that opposing transport systems produced no net accumulation of CoQ_10_ in the brain. This was observed for both the PBEC model and for a mouse BBB cell line, bEnd.3 [141]. Transporters at the BBB generally act together to limit systemic lipoprotein and cholesterol transfer to the brain.

Based on known lipoprotein transporters at the BBB, pharmacological inhibitors were screened for their effect on CoQ_10_ transport across bEND.3 cells. BLT-1 is an inhibitor that targets the scavenger receptor (SR-B1), which interacts with HDL on the apical membrane to execute caveolae-mediated lipoprotein influx [142]. RAP was used to inhibit the low-density lipoprotein receptor (LDLR) family members, including the LDL receptor-related protein 1 (LRP-1). LRP-1 interacts with a variety of ligands to mediate BBB integrity [143]. FPS-ZM1 was used to inhibit the receptor for advanced glycation end products (RAGE). RAGE mediates the influx and opposes LRP-1-mediated efflux of ApoE and amyloid-beta across the BBB [144]. Upon inhibition of either SR-B1 or RAGE, apical-to-basal transport of CoQ_10_ was decreased to 44% and 50% of the control, respectively, whereas basal-to-apical transport exhibited no change [141]. The authors suggested a direction-specific role of both receptors for CoQ_10_ transcytosis into the brain. In contrast, inhibition of LRP-1 (and LDLR family members) increased apical-to-basal CoQ_10_ transport by 168%, with no significant change in the reverse transport, suggesting that LRP-1 directs CoQ_10_ transport out of the brain [141].

Wainwright et al. used pABA, a competitive inhibitor of polyprenyldiphosphate-4-hydroxybenzoate transferase (Coq2), to create a CoQ_10_-deficient model of the BBB. Treatment with pABA significantly decreased cellular CoQ_10_ content and mitochondrial respiratory chain complex activities in bEND.3 cells [141]. Cells treated with *p*ABA also had reduced tight junction integrity and increased permeability (paracellular transport, Figure 6b), causing a shift in the net transport of CoQ_10_ in the blood-to-brain direction. Under these pABA-inhibited conditions, the RAGE receptor was responsible for some of the net CoQ_10_ transport, while the SR-B1 receptor-mediated transport was no longer active. However, the use of pABA to inhibit CoQ_10_ biosynthesis has possible confounding effects. pABA is a substrate for Coq2, and the resulting 4-amino-3-polyprenylbenzoic acid may exert negative effects on cell endocytic trafficking (see Section 2.1). Studies with mammalian cultured cells show that once pABA is prenylated, it is converted to a redox-active “dead-end” amino-containing analog of DMQ, which may interfere with respiratory electron transport [47]. Instead of pABA, it has been suggested that 4-nitrobenzoic acid (4-NB) should be used as a competitive inhibitor of Coq2, since it is not prenylated [47].

Interestingly, co-supplementation of CoQ_10_ with α-tocopherol or Trolox, a water-soluble analog of α-tocopherol that does not interact with lipoproteins, in *p*ABA-treated and untreated bEND.3 cells caused an increase in CoQ_10_ efflux in the brain-to-blood direction. This result suggests that the co-supplementation of CoQ_10_ with α-tocopherol for individuals with CoQ_10_ deficiency might decrease CoQ_10_ delivery to the brain [141]. More work is needed to determine whether this effect is due to excessive antioxidants blocking uptake of lipoproteins containing CoQ_10_ or if mechanisms of CoQ_10_ uptake are mediated by changes in oxidative stress.

Finally, ABCB1 (P-glycoprotein) was studied since it is reported to decrease CoQ_10_ transport across the CaCo-2 intestinal epithelial-barrier model [145]. Inhibition of P-glycoprotein with verapamil had no effect on CoQ_10_ transport across the BBB, implying a mechanistic difference between exogenous CoQ_10_ uptake in small intestinal epithelial cells versus BBB endothelial cells [141]. However, Wainwright et al. note that in the experiments by Itagaki et al. [145] with CaCo-2, exogenous CoQ_10_ was supplemented in its free, un-micellized form, which is not favorable for small intestine uptake CoQ_10_. The existence of brain endothelial membrane receptors that enhance CoQ_10_ transcytosis in a selective direction across the BBB opens the possibility for antagonistic and/or agonistic supplements to increase exogenous CoQ_10_ delivery to the brain.

CD36 is another receptor that has been identified in the uptake of exogenous CoQ. CD36 is a member of the SR family of transmembrane glycoproteins and a class-B scavenger receptor that exhibits high structural and functional homology to SR-B1 [146]. CD36 transports fatty acids and plays a regulatory role in lipid metabolism [147]. HEK293 human embryonic kidney cells overexpressing CD36 took up 57% more CoQ_9_ than wild-type cells [148]. Importantly, receptor-mediated uptake of exogenous CoQ_9_ and CoQ_10_ was observed only when solubilized with Intralipid^®^, an emulsion of soybean oil, glycerol, and egg yolk phospholipids. Receptor-mediated uptake was not reported for solvent- or detergent-solubilized CoQ [148]. Additional work by Anderson et al. [148] reports that CD36 drives CoQ uptake in brown adipose tissue (BAT) and is required for normal BAT function (see Section 4.1).

#### 3.2.2. Saposin B Binds CoQ_10_ and Regulates CoQ_10_ Content

Prosaposin is a precursor protein that undergoes proteolytic cleavage to generate four small proteins known as saposins (saposins A, B, C, and D). Saposin B is a heat-stable glycoprotein that is detected in many tissues. It forms shell-like dimers to bind sphingolipids and glycerol-phospholipids, and primarily localizes to lysosomes [149,150,151]. CoQ_10_ bound to saposin B was first identified in protein fractions collected from human urine samples and later detected in cell-lysates from HepG2 hepatocytes and human sperm [65]. CoQ_10_ was shown to be transferred to erythrocyte ghost membranes when incubated with saposin B-CoQ_10_ complexes at pH 4.5, suggesting that saposin B may play a role in delivery of CoQ_10_ to acidic compartments, such as the lysosome [65].

Saposin B has high binding affinity for CoQ_10_ as well as CoQ_9_ and CoQ_7_ isoforms, but a low affinity for α-tocopherol at pH 7.4 [65]. His-tagged saposin B expressed in *E. coli* was also shown to bind CoQ_10_ in in vitro experiments in a pH-dependent manner, and showed that glycosylation of saposin B is not essential for CoQ_10_ binding [152]. Saposin B has a comparable affinity for γ-tocopherol and CoQ_10_ at neutral and basic conditions, but a higher affinity for γ-tocopherol at acidic conditions, suggesting that the binding affinity of saposin B for γ-tocopherol is greater than that of CoQ_10_ [64]. In binding competition assays, two lipids in hexanes were mixed with aqueous saposin B at pH 7.4 and the resulting aqueous saposin B-bound lipids were measured. Interestingly, saposin B bound γ-tocopherol over CoQ_10_ despite similar binding affinities and bound α-tocopherol over CoQ_10_ despite lower binding affinity [64]. Based on known interactions of saposin B dimers with phospholipids, the authors postulated that saposin B dimers could bind two molecules of CoQ_10_ or one molecule of CoQ_10_ and two molecules of tocopherol. Their results suggest that the binding of one molecule of CoQ_10_ may accelerate the binding of two molecules of α-tocopherol. The crystal structure of saposin B binding chloroquine was later resolved [153] and molecular docking studies identified two binding sites that are accessible to a variety of ligands [154].

Mutations in prosaposin and changes to its cellular content have been reported to affect CoQ_10_ levels and mitochondrial function. Knockdown of prosaposin or overexpression of a prosaposin mutant that cannot form saposin B in HepG2 cells causes a significant reduction in both whole-cell and mitochondrial CoQ_10_ content [155]. Likewise, overexpression of prosaposin leads to an increase in whole-cell and mitochondrial CoQ_10_ content, suggesting that prosaposin and/or saposin B plays a role in regulating CoQ_10_ levels [155]. Knockdown of prosaposin in CaCo-2 cells also decreases cellular CoQ_10_ content, causing a decrease in ATP formation and glucose consumption [156]. As observed previously, a decrease in ATP formation is associated with disruption in tight junction formation and increased permeability.

More recent work has modeled long-term CoQ_10_ deficiency in HepG2 cells and found that decreased CoQ_10_ content was associated with decreased prosaposin levels [157]. Treatment with 4-NB decreased CoQ_10_ cellular content after three days and decreased prosaposin protein levels after three months. Decreased CoQ_10_ content and prosaposin protein levels were observed up to sixteen months with 4-NB treatment. The addition of the CoQ precursor 4-HB to cells treated with 4-NB significantly rescued both CoQ_10_ content and prosaposin protein levels. Studies in prosaposin-deficient mice have not investigated CoQ content, though these mice often die one to two days after birth [158]. In summary, the disruption of prosaposin or saposin B decreases CoQ_10_ content, indicating that saposin B is a CoQ_10_ transporter that plays a role in regulating CoQ_10_ levels.

## 4. Rodent Models

*Mus musculus* (mice), *Rattus norvegicus* (rats), and *Cavia porcellus* (guinea pigs) are common rodent models for the study of CoQ biosynthesis and uptake. Both mice and rats endogenously synthesize CoQ_9_ as the predominant CoQ isoform, with CoQ_10_ present as a minor isoform. This is in contrast to humans, guinea pigs, and other mammalian models, where CoQ_10_ is the major isoform [8]. Rodent models are also used to study the effect of genetic mutations on endogenous CoQ biosynthesis [159]. Hence, rodent models have been used to assess how CoQ supplements affect the distribution and content of CoQ in tissues in both normal animals as well as those harboring gene mutations causing primary CoQ deficiency [6,160].

### 4.1. Organism-Level Uptake of CoQ

Due to their insolubility in water CoQ_9_ and CoQ_10_ are poorly bioavailable, necessitating unusually high doses [2,7]. Dosing in humans averages around 200 mg of CoQ_10_ taken twice daily. However, when used to treat neurodegenerative diseases, doses can reach up to 3000 mg per day [161]. As a high-molecular-weight, lipophilic molecule, CoQ_10_ is poorly absorbed by the gastrointestinal (GI) tract [162]. However, the consumption of CoQ_10_ with other dietary lipids, as well as various supplement formulations, can enhance its uptake [7,162].

A majority of rodent studies investigate the effects of CoQ_10_ supplements via oral treatment. The addition of CoQ_10_ to rodent chow is a common feeding method. Similar to studies in human patients, high doses have been used to study CoQ_10_ uptake in mice and rats [162,163]. As the CoQ_10_-supplemented chow is digested, the hydrophobic CoQ_10_ is taken up at the GI tract. In male Sprague Dawley rats, CoQ_10_ permeability along the GI tract was investigated using a side-by-side diffusion apparatus, in which isolated GI tract tissues were placed in between a CoQ_10_-containing donor medium and a receptor chamber [164]. These studies indicated that the greatest permeability is located at the duodenum, followed by the colon and the ileum.

Within the intestinal enterocytes, absorbed CoQ_10_ is likely to be packaged along with other dietary lipids into chylomicrons, which are transported into the lymphatic system [165]. Chylomicrons in the lymph are then delivered to systemic circulation via the thoracic duct and subjected to processing by lipoprotein lipase in the capillaries. The resulting chylomicron remnants are taken up by the liver (Figure 7) [166]. This delivery may be responsible for CoQ_10_ enrichment seen in the liver. We note here the assumption that orally supplemented CoQ_10_ is trafficked in a manner similar to other dietary lipids such as vitamin E [167] and cholesterol [168]. However, there is a lack of direct experimental support for the presence of supplemented CoQ_10_ in chylomicron remnants.

In the liver, CoQ_10_ derived from dietary sources is likely to be combined with de novo synthesized CoQ where it would be assembled into lipoprotein particles, with the majority in very low-density lipoprotein (VLDL) [164]. VLDL secreted by the liver is processed by lipoprotein lipase in the capillaries of various tissues to form intermediate-density lipoprotein and low-density lipoprotein (LDL) [168]. Lipoprotein particles extracted from human plasma find LDL to be most enriched in CoQ_10_ content [169]. CoQ_10_H_2_ is an important antioxidant in circulating lipoproteins and enhances the resistance of LDL to lipid oxidation [170,171].

High-density lipoprotein particles (HDL) isolated from human plasma contain approximately 50% of the CoQ_10_ content as compared to LDL [169]. This relationship between LDL and HDL CoQ content is also observed in mice [148]. In a mouse model, the absence of apolipoprotein ApoA1, a major component of HDL, decreased mitochondrial cardiac CoQ content by 75% [172]. Following ischemia/reperfusion of the left ascending coronary artery, the infarct size increased significantly in the homozygous ApoA1^−/−^ mice, as compared to the heterozygous ApoA1^−/+^ mice, which in turn had larger infarcts than the parental control (WT) mice [172]. Although oral supplementation of CoQ_10_ in ApoA1^−/−^ mice did not affect the cardiac CoQ_10_ content, intraperitoneal (IP) injection of CoQ_10_ restored the cardiac mitochondrial CoQ_10_ content of the ApoA1^−/−^ mice to WT levels, and decreased their infarct size to that of WT mice following ischemia-reperfusion [172]. These results suggest a possible role for ApoA1 in the uptake and trafficking of CoQ_10_ from the GI tract to the heart; however, further study is needed to confirm this relationship. It is unfortunate that in this study the authors did not quantify the endogenous CoQ_9_ content as it may have further informed the outcome of supplementation with CoQ_10_. However, another confounding factor is that the authors used CoQ_9_ as an internal standard for the quantification of CoQ_10_. This is problematic for two reasons: (1) mouse heart contains endogenous CoQ_9_, and (2) commercially available standards of CoQ_9_ can be contaminated with CoQ_10_.

It is important to note that many studies in mice and rats utilize CoQ_10_ as the supplemented isoform, although the predominant endogenous isoform is CoQ_9_. This may generate conflicting outcomes when comparing rats or mice with guinea pigs, which solely produce CoQ_10_ as the endogenous isoform. Several studies in both rats and mice have shown that supplementation of CoQ_10_ increases endogenous CoQ_9_ levels in various tissues, serum, and mitochondria [163,173,174]. This is likely due to a sparing effect of supplemented CoQ_10_ on the endogenous CoQ_9_ pool in which the addition of CoQ_10_ to the cellular CoQ pool “spares” the turnover of CoQ_9_; a similar phenomenon is observed when CoQ_10_ supplements preserve levels of α-tocopherol or the tissue content of CoQ_9_ [175,176]. The cardioprotective effects of CoQ_9_ and CoQ_10_ supplementation were investigated in guinea pigs [177]. CoQ_9_ and CoQ_10_ were found to have functionally identical effects, as both isoforms conferred comparable cardioprotection “as evidenced from the comparable degree of the postischemic ventricular recovery and reduction in myocardial infarct size and cardiomyocyte apoptosis”. Lekli et al. further claimed that CoQ_9_ can be converted to CoQ_10_ after supplementation, although no metabolic labeling of the fed CoQ_9_ was performed to enable a more substantive conclusion. It is more likely the elevation in CoQ_10_ content was due to the aforementioned sparing effect.

CoQ_10_ present in lipoproteins may be delivered to cells via receptor-mediated uptake, to facilitate the entry of CoQ at the cellular level [2]. A specific example is the CD36 scavenger receptor-mediated uptake of exogenously supplied CoQ_10_ into mouse brown adipose tissue (BAT) [148]. CD36 is present in many tissues, with the highest abundance in adipose tissue, heart, lung, and muscle, and less expression in spleen and liver [148]. BAT functions in non-shivering thermogenesis, a process generated by uncoupled respiration [178]. To perform this function, BAT depends on the uptake of CoQ rather than through its own de novo biosynthesis [148]. BAT isolated from mice lacking the CD36 receptor (CD36^−/−^), have significantly reduced endogenous CoQ levels. Uptake of Intralipid^®^-solubilized CoQ_10_ in CD36^−/−^ mice via intraperitoneal injection is significantly reduced in BAT compared to wild-type mice, while no difference was observed in liver or muscle tissues. Moreover, CD36^−/−^ mice show a significant increase in CoQ_10_ serum levels after intraperitoneal injection of exogenous CoQ_10_ when compared to wild-type mice, suggesting that BAT is a major sink for exogenous CoQ. CD36^−/−^ BAT mitochondria showed impaired substrate utilization and decreased basal rates of respiration, which was rescued upon the addition of CoQ_2_. CD36 is found to be exclusively on the plasma membrane in BAT, and BAT lacking CD36 lacks the characteristic browning of BAT following cold exposure. Likewise, CD36^−/−^ mice exhibit defective non-shivering thermogenesis. The mechanism by which CD36 mediates the uptake of CoQ_9_ or CoQ_10_ into BAT is not clear. The authors suggested that uptake could proceed either via CD36-mediated endocytosis of lipoprotein particles or that CoQ could be selectively taken up from a bound lipoprotein [148]. In either case, these findings provide evidence for receptor-mediated uptake of CoQ. (See Section 3.2.1 for further discussion of receptor-mediated uptake of CoQ_10_ at the blood–brain barrier.)

Extensive reviews and literature compare the efficacy of supplementation with CoQ_10_ and CoQ_10_H_2_ supplementation in humans [2,7,179]. However, few studies investigate the relative bioavailability of CoQ_10_H_2_ relative to CoQ_10_ in a rodent model. In rats, CoQ_10_ is converted into CoQ_10_H_2_ during or after absorption by the intestine [162], yet exogenous supplementation with CoQ_10_H_2_ has been posited to increase the molecule’s bioavailability. Mice lacking the ability to synthesize CoQ due to homozygous partial loss of function mutations in *Coq9* (*Coq9*^−/−^) were supplemented with water-soluble formulations of both CoQ_10_ and CoQ_10_H_2_ [180]. While both formulations increased tissue levels relative to the Coq9^−/−^ control, the tissues of mice fed CoQ_10_H_2_ tended to have greater enrichment in CoQ_10_ compared to mice fed CoQ_10_ [180]. Measurement of cerebrum mitochondrial CoQ_10_ content revealed the same trends. A recent study used positron emission tomography imaging to detect the uptake and distribution of ^11^C-CoQ_10_ and ^11^C-CoQ_10_H_2_ that were administered to rats via tail vein injection [181]. The uptake of the labeled CoQ_10_H_2_ in the cerebrum, cerebellum, white adipose tissue, muscle, kidney, testis, and BAT was higher than that of the labeled CoQ_10_. However, uptake of the ^11^C-labeled CoQ_10_H_2_ was lower in the spleen as compared to labeled CoQ_10_. Although the delivery of CoQ_10_H_2_ in this ^11^C-labeling study was via intravenous injection [181], it is interesting that the enrichment achieved by CoQ_10_H_2_ in brain tissues was also observed for diet-delivered CoQ_10_H_2_ in the *Coq9*^−/−^ mouse study [180].

### 4.2. Tissue Localization of Supplemented CoQ

Understanding the localization of supplemented CoQ to specific tissues may be important to treating primary CoQ deficiencies that affect specific or multiple organs. However, as discussed above, the enrichment of CoQ in specific tissues of rats and mice can be affected by the redox state of the supplement. There is also variability observed in rats, mice, and guinea pigs following supplementation with different formulations of CoQ. This variability can be attributed to the conditions of supplementation including dose, mode of supplement administration, lipid extraction, and quantification methods used, as well as the genetic background or strain of the animal model.

#### 4.2.1. Rat

IP injection into rats of ^3^H-CoQ_10_, labeled at the isoprenyl methylene closest to the benzoquinone ring, permitted quick evaluation and extensive uptake into a number of tissues. Enriched tissues include the liver, spleen, adrenals, and ovaries, with limited uptake into the heart and thymus, as well as white blood cells [182]. This is in contrast to the poor dietary uptake of ^3^H-CoQ_10_. Over a two-week period of oral supplementation, only 4–6% was taken up while the rest was excreted as radioactive metabolites in urine and feces. Analysis of the excreta revealed that half of the radioactive content excreted was present as non-metabolized CoQ_10_ in feces, suggesting that excess CoQ_10_ is removed through bile excretion (Figure 7) [182]. The results concur with prior studies that showed both oral and IP injections of CoQ_10_ resulted in liver and spleen enrichment, while none was detected in the kidney, heart, or brain tissue [173,183]. However, Kwong et al. [174] observed modest CoQ_10_ enrichment of the heart and kidney tissues. Differences in enrichment between studies may be attributed to dosage and method of feeding. Reahal and Wrigglesworth and Lonnrot et al. supplemented CoQ_10_ in suspension [173,183], whereas Kwong et al. appear to supplement the chow with powdered CoQ_10_ [174]. CoQ_10_ is better taken up as an emulsion compared to a powder [2,7].

Intravenous (IV) administration in rats of 4 mg/kg CoQ_10_ solubilized in lipid microspheres or chromophore EL has also been investigated [184]. Chromophore EL was described as a polyethoxylated castor oil that contained approximately 40 units of ethylene oxide to each triglyceride unit along with the addition of undefined antioxidants [185]. Results from Scalori et al. [184] indicate that CoQ_10_, whether orally or IV administered, leaves the blood plasma quickly and is particularly enriched in the liver, although with varying kinetic profiles. The heart and kidneys of these rats were also enriched in CoQ_10_. However, smaller doses of CoQ_10_ were needed to achieve the same effect when supplemented by IV [184].

Topical application of CoQ_10_, a lesser-studied method of administration, has also been found to increase skin concentrations in Sprague Dawley rats [184]. This is consistent with more recent data indicating that topical administration of CoQ_10_ on the forearms of human females significantly increased levels at the skin surface and the deeper epidermis layer [186], and reduced UVB-induced wrinkle formation [187].

#### 4.2.2. Mouse

Similar results for tissue enrichment of CoQ are seen in mouse models. Following a 13-week regimen of CoQ_10_ supplementation in a soybean oil suspension, the serum and liver homogenate had increased levels of CoQ_10_ [175]. No changes to CoQ_10_ content were observed in the brain, heart, kidney, or skeletal muscle homogenates. However, a higher dose, 200 mg/kg/day, and a longer regimen of CoQ_10_ were shown to increase cerebral cortex levels of CoQ_10_, CoQ_10_H_2_, and CoQ_9_H_2_ [188]. These results are recapitulated in a study looking at the short-term administration of CoQ_10_H_2_ [189]. Imaging mass spectrometry also shows increased CoQ_10_H_2_ in several regions of the brain, including the hippocampus, cerebellum, and brain stem [189]. Matthews et al. showed oral supplementation of CoQ_10_ exerted neuroprotective effects by reducing striated lesion volumes produced from systematic administration of 3-nitropropionic acid, a complex II inhibitor [188].

#### 4.2.3. Guinea Pig

In the guinea pig model, Passi et al. [190] investigated the effect of CoQ_10_ and ubiquinol_10_-diacetate (CoQ_10_-DIA) provided in the diet, on liver subcellular fraction content. CoQ_10_H_2_ modified by acetate at the ring 1,4 positions, (CoQ_10_-DIA) is readily hydrolyzed and taken up as CoQ_10_H_2_. The study found that both CoQ_10_ and CoQ_10_-DIA taken up were mainly enriched in the heavy mitochondria and light mitochondria/lysosome fractions. Modest increases in CoQ_10_ content were also seen in the nucleus/debris and microsome fractions. There were no significant differences in subcellular enrichment between CoQ_10_ and CoQ_10_-DIA [190].

Across all three rodent models, exogenously supplemented CoQ_10_ is enriched mainly in the liver, plasma, and serum. While there remains conflicting data on CoQ enrichment in other tissues, it appears that high dosage, IV, or IP administration may be able to overcome certain barriers to uptake into tissues such as the brain, heart, and kidney.

### 4.3. Subcellular Localization of Supplemented CoQ

When the CoQ sequestered in lipoproteins is taken up by a cell, it must be trafficked to intracellular membranes. Currently, little is known about the mechanisms of intracellular trafficking used to shuttle CoQ. However, the subcellular enrichment of CoQ following supplementation has been studied in mice, rats, and guinea pigs.

#### 4.3.1. Rat

Under normal physiological conditions, endogenous CoQ_9_ in rat liver tissue is predominantly located in the mitochondrial subcellular fraction [183]. After CoQ_10_ supplementation, subcellular fractionation of hepatic tissue cells was investigated [191]. The pellet containing lysosomes and light mitochondria was especially enriched in CoQ_10_, with no change to the endogenous levels of CoQ_9_. Bentinger et al. [182] conducted an investigation into the subcellular localization of ^3^H-CoQ_10_ in the liver, finding that nearly 60% of radiolabeled CoQ_10_ localized to the lysosome after IP supplementation, whereas the mitochondrial fractions contained only approximately 11% of the total ^3^H-CoQ_10_ recovered [182]. This is consistent with a prior study, where IP administered CoQ_10_ was primarily localized to the soluble cytosolic fraction of liver cells and was recovered as the oxidized form, not the reduced form [183]. As CoQ_10_ is highly hydrophobic, it is unclear why the concentration was found to be greatest in the soluble cytosolic fraction. It is also worth mentioning that this study does not indicate precautions taken to mitigate autoxidation of reduced CoQ_10_H_2_ in liver extracts during tissue homogenization and lipid extraction. Such precautions may include quick procedural timing, the use of acidified solvents during extraction, and inert gases to prevent autoxidation. An LC–MS/MS quantification method sensitive to the CoQ redox state was recently developed to minimize measurement error due to autoxidation [119].

In a separate rat study, the brain and brain mitochondria were shown to be enriched in CoQ_10_ following a two-month supplementation regimen at a dose of 200 mg CoQ_10_/kg per day [188]. However, the CoQ_9_ content was not significantly increased in this study. Another study administering 150 mg/kg CoQ_10_ per day for 13 weeks demonstrated an increase in heart, skeletal muscle, liver, and kidney mitochondrial CoQ_10_ content, as well as an additional increase in CoQ_9_ content in mitochondria of heart, skeletal muscle, and kidney tissue homogenate [174]. Only the mitochondria-enriched fractions were examined for CoQ_9_ and CoQ_10_ contents in these two studies.

#### 4.3.2. Mouse

Mice supplemented with CoQ_10_ generally presented with increases in both CoQ_9_ and CoQ_10_ content in the liver, heart, kidney, and skeletal muscle tissue mitochondria, but not in the brain [163,176]. The data obtained by Sohal et al. were subjected to further analyses to consider the length of treatment and dosage factors. General trends indicated that mice treated for 17.5 months with CoQ_10_ had larger magnitude increases in CoQ content as compared to mice treated for 3.5 months [163]. No consistent dose-dependent trend was reported. However, these conclusions are solely based on the examination of mitochondria-enriched fractions

#### 4.3.3. Guinea Pig

In the guinea pig model, Passi et al. [190] investigated the effect of dietary CoQ_10_ and CoQ_10_-DIA on liver subcellular fraction content. The authors found that both CoQ_10_ and CoQ_10_-DIA taken up were mainly enriched in the heavy mitochondria and light mitochondria/lysosome fractions. Modest increases in CoQ_10_ content were also seen in the nucleus/debris and microsome fractions. There were no significant differences in subcellular enrichment between CoQ_10_ and CoQ_10_-DIA.

### 4.4. Intracellular Trafficking of CoQ

There is little information on the intracellular trafficking of exogenously supplemented or endogenously synthesized CoQ in rodent models. Data gleaned from other cellular models, including *Saccharomyces cerevisiae* and human cell lines, can inform such studies. Intriguingly, ADCK2, the mouse homolog of yeast Cqd1 (discussed in Section 2.3), has been shown to play a role in modulating the transport of lipids into mitochondria for fatty acid β-oxidation and CoQ biosynthesis [95]. ADCK2 and Cqd1 are members of the UbiB/Coq8 family of atypical kinases. *Adck2*^+/−^ mice exhibit haploinsufficiency, with CoQ deficiency specifically in skeletal muscle and defects in mitochondrial fatty acid oxidation [95]. CoQ_10_ supplementation increased the content of CoQ_10_ in the skeletal muscle of the *Adck2*^+/−^ mice, enhanced physical performance, and decreased plasma lactate levels [95]. It seems likely that ADCK2 acts similarly to Cqd1, the yeast ortholog, and is necessary to retain CoQ within mitochondria [93].

### 4.5. Mouse Models of CoQ Deficiency and the Effects of CoQ Supplementation

Investigation of human mutations in *COQ* genes has been studied in transgenic mouse models to assess the organism-level effect of canonical single nucleotide variants. Transgenic mice with homologous gene KOs have also been used to assess and investigate phenotypic consequences, and to evaluate the efficacy of treatment with CoQ_10_ supplements.

Saiki et al. [192] investigated the efficacy of CoQ_10_ supplementation in the *Pdss2^kd/kd^* (kidney disease) mouse. The *Pdss2^kd/kd^* mouse harbors a spontaneous homozygous V117M mutation resulting in a partial loss of function in PDSS2, one of the two subunits of polyprenyldiphosphate synthase essential for CoQ biosynthesis, and a homolog of yeast Coq1 [192,193]. *Pdss2^kd/kd^* mice present with CoQ deficiency in early adulthood, resulting in a kidney disease similar to the collapsing glomerulopathy variant of focal segmental glomerulosclerosis [192]. Mice harboring a podocyte-specific loss of function knockout of *Pdss2* reproduced a more severe phenotype of the *Pdss2^kd/kd^* mice, indicating that the kidney disease in the *Pdss2^kd/kd^* mouse results from the CoQ deficiency in the podocytes [193]. Intriguingly, supplementation with CoQ_10_ afforded a dramatic rescue of the kidney disease as ascertained by a 5-fold reduction in urine albumin content. However, CoQ_10_ supplementation, even at doses of 200 mg/kg per day did not affect the kidney CoQ_10_ content as compared to the untreated *Pdss2^kd/kd^* mouse. It was speculated that the CoQ_10_ supplementation may exert its effects via the filtration of blood plasma components [194]. The podocyte foot processes act as a filtration barrier, and the CoQ_10_ may function primarily as a membrane antioxidant [192]. A more recent study using the same *Pdss2^kd/kd^* mouse model supports CoQ_10_ supplementation (0.5% CoQ_10_ supplemented food), as a preventative treatment for the progression of nephrotic syndrome induced by deficiency [138]. Kidney CoQ_9_ and CoQ_10_ levels were increased, and damage indicated by renal histology is significantly dampened after six months of supplementation with CoQ_10_. CoQ_10_ supplementation rescued defects in the oxidation of sulfide, and decreased oxidative stress in the kidneys of *Pdss2^kd/kd^* mice [138].

The kidneys of *Pdss2^kd/kd^* mice are shown to have a defect in the sulfide oxidation pathway resulting in increased hydrogen sulfide and depleted glutathione [195]. Deficiencies in CoQ impede the sulfide oxidation pathway by decreasing SQR activity [195,196]. Further investigation of mouse *Coq9* mutants revealed similar results. *Coq9^R239X^* mice, with 10-15% residual CoQ_9_ content, had significantly less SQR levels in the kidneys and cerebrum whereas *Coq9^Q95X^*, with 40–50% residual CoQ_9_ content in the same tissues, had no decrease in these tissues [196]. Both models exhibit 10–20% residual CoQ_9_ content in muscle and have decreased SQR in this tissue. While low CoQ content disrupts sulfide oxidation, supplementation with 100 μM CoQ_10_ upregulates the sulfide oxidation pathway, restoring its function as well as other downstream and tangential effects of impaired sulfur metabolism [197].

Investigation of the mouse *Coq7* homolog, *Mclk1*, showed that homozygous knockout resulted in embryonic lethality [198]. Heterozygous *Mclk1^+/^*^−^ mice, however, live a normal lifespan with no significant defects in subcellular CoQ_9_ levels or mitochondrial morphology. Subcellular distribution of CoQ_9_ is altered, which may explain the lowered oxygen consumption and ATP production phenotype of heterozygous *Mclk1^+/^*^−^ mitochondria [199,200]. Supplementation with CoQ_10_ significantly increases mitochondrial CoQ_10_ content and restores the electron transport chain defect [198].

Rare deleterious mutations in the human *COQ9* gene result in severe multisystem disorders, affecting the central nervous system, cardiac, and renal systems [201]. Homozygous *Coq9* mice (*Coq9^R239X/R239X^*) were prepared to model the human COQ9^R244X^ deficiency [180]. These mice exhibit profound brain damage due to oxidative damage, neuronal apoptosis, and demyelination in the pons and medulla oblongata [180]. After two months of 240 mg/kg/day CoQ_10_ or CoQ_10_H_2_ oral supplementation, significant increases were detected in plasma, liver, and muscle tissue, followed by smaller increases in the heart, kidney, cerebrum, and cerebellum [180]. Supplementation with CoQ_10_H_2_ appeared to alleviate histopathological symptoms of CoQ deficiency in the brain, whereas CoQ_10_ did so to a lesser extent.

Such dramatic effects of CoQ_10_ supplementation have also been seen in some human patients harboring primary CoQ deficiency [202,203,204,205]. The parallels between the rodent disease models and the human clinical outcomes emphasize the benefit of CoQ-deficient mouse research models.

## 5. Conclusions

CoQ is an essential and ubiquitous lipid with many complex roles. However, there remains much to be learned about the mechanisms responsible for its uptake, intracellular trafficking, and distribution. Here, we have summarized the work in several model organisms used to investigate how CoQ is trafficked.

Yeast is an extraordinarily powerful model organism that has been used to identify proteins involved in the uptake and trafficking of exogenous CoQ_6_ to mitochondrial respiratory complexes. The yeast *coq* mutants completely lack CoQ_6_ biosynthesis and are thus respiratory deficient and fail to grow in a medium containing a non-fermentable carbon source. Supplementation with exogenous CoQ restores respiration and growth of *coq* mutants, and supplementation with the soluble CoQ_2_ analog can bypass defects in endocytosis and membrane trafficking [18,19,58]. Fernandez-del-Rio et al. [18] identified six genes that are essential for the uptake and trafficking of exogenous CoQ_6_ to mitochondrial respiratory complexes—*RTS1*, *CDC10*, *RVS161*, *RVS167*, *VPS1*, and *NAT3*. More work is needed to determine the role that each of these genes plays in the uptake and trafficking of exogenous CoQ_6_. While supplementation of *ORFΔcoq2Δ* double mutants with exogenous CoQ_6_ is a powerful approach for identifying genes required for CoQ_6_ uptake and trafficking, genes that are redundant or essential for respiration cannot be identified with exogenous CoQ_6_ rescue screens.

Yeast is also a powerful model for determining pathways and proteins responsible for the trafficking of endogenously synthesized CoQ_6_. Biosynthesis and transport of endogenously synthesized CoQ_6_ from the mitochondria in yeast is possibly mediated by ERMES [88,89]. However, direct transport of CoQ_6_ through ERMES has not yet been observed. Cqd1 and Cqd2 are mitochondrial inner membrane proteins that face the intermembrane space and influence the cellular distribution of endogenous CoQ_6_, and likely play a role in transporting CoQ_6_ from the mitochondrial inner membrane to the outer membrane and ER [93]. It is likely that other still unidentified CoQ_6_ transporters may contribute to trafficking.

Mammalian cell lines are an ideal model for investigating the cellular uptake and intracellular distribution of CoQ. Synthetic vehicles, such as cyclodextrin, caspofungin, micelles, liposomes, and nanodisks have been shown to dramatically improve the bioavailability of exogenous CoQ [117]. Moreover, supplementation with CoQ_10_H_2_ displays improved bioavailability in human dermal fibroblasts compared to CoQ_10_ [135], a trend that is also observed in rodent models [180,181]. Treatment of HL-60 cells with brefeldin-A shows that intracellular distribution of both endogenous and exogenous CoQ relies at least in part on the endomembrane system [83], though more work is needed to identify essential players of this pathway for CoQ transport. Recent work in HeLa cells has identified STARD7 as a phosphatidyl transfer protein involved in the export of endogenous CoQ from the mitochondria to the plasma membrane, where it plays a role in ferroptosis suppression [63]. Work in mouse endothelial brain cells has identified scavenger receptor SR-B1, receptor for advanced glycation endproducts (RAGE), and LRP-1 to be involved in transport of lipoprotein-associated CoQ_10_ across the blood–brain barrier [141]. CoQ_10_ may also be delivered to BAT cells by receptor-mediated uptake, via-the CD36 scavenger receptor [2,148].

Rodent models are useful for investigating the bioavailability of CoQ supplements and CoQ distribution in tissues of both wild-type animals as well as those harboring gene mutations that cause primary CoQ_10_ deficiency [6,160]. Dietary CoQ in rodents is taken up at the GI tract [164], and is then likely packaged into chylomicrons by intestinal enterocytes and transported into the lymphatic system, though more work needs to be performed to identify the pathways of endothelial uptake and to detect the supplemented CoQ in chylomicrons. CoQ from chylomicron remnants is likely repackaged with de novo CoQ at the liver and secreted in lipoprotein particles, with the majority in VLDL. Tissue localization of supplemented CoQ in rodents varies across studies and CoQ formulations, but enrichment is usually observed in the liver, plasma, and serum, with varying results in other tissues such as the heart, brain, and kidney. Supplemented CoQ is enriched in subcellular mitochondrial and light mitochondrial/lysosomal fractions. Moreover, transgenic mouse models have been used to investigate human mutations and canonical single nucleotide variants in *Coq* genes in order to evaluate the efficacy of treatment of CoQ-deficient mouse models with CoQ_10_ supplements.

It is important to note that comparisons across studies are challenging due to variability in CoQ formulations and dosage. Comparative studies of CoQ_10_ and CoQ_10_H_2_ must use proper precautions and controls to prevent rapid oxidation of CoQ_10_H_2_. Unfortunately, rodent studies that investigate CoQ_10_ uptake sometimes overlook endogenous CoQ_9_ content, despite CoQ_9_ being the predominant isoform in mice and rats. Many aspects of CoQ uptake and transport remain unknown or poorly understood. What is the relative importance of the endomembrane system and membrane contact sites in the transport of CoQ? How are CoQ transport and intracellular distribution of CoQ regulated? Elucidating the pathways of how organisms assimilate CoQ supplements, its cellular uptake, and pathways of intracellular distribution is important to understanding the functional roles of CoQ and may lead to improved therapeutics for exogenous supplementation.

## Figures and Tables

**Figure 1 antioxidants-12-01391-f001:**
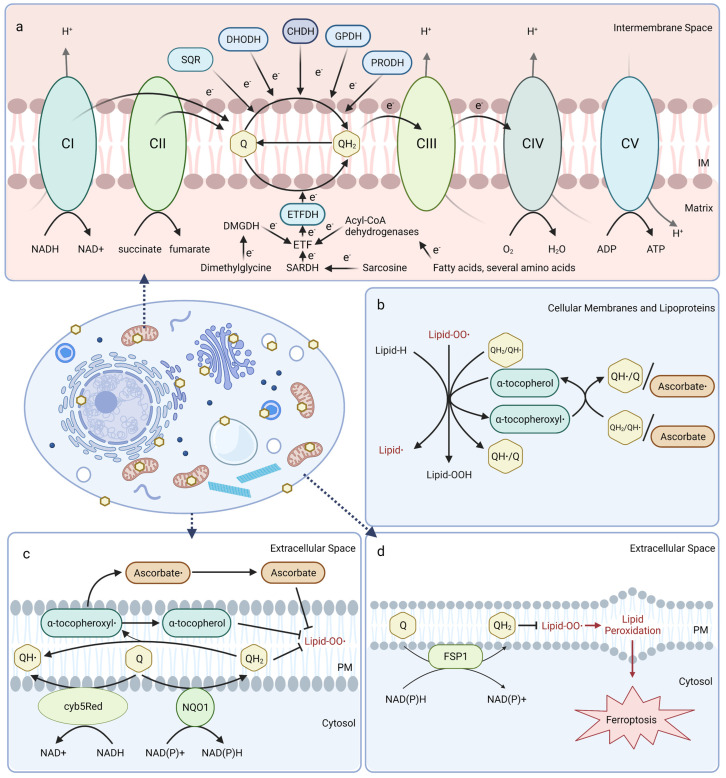
Coenzyme Q functions. (**a**) Mitochondrial CoQ shuttles electrons in the electron transport chain and is reduced by dehydrogenases in diverse metabolic pathways. CI, Complex I; CII, Complex II; CIII, Complex III; CIV, Complex IV; CV, Complex V; DHODH, dihydroorotate dehydrogenase; GPDH, glyceraldehyde-3-phosphate dehydrogenase; SQR, sulfide:quinone oxidoreductase; CHDH, choline dehydrogenase; PRDH, proline dehydrogenase; ETF, electron transfer flavoprotein; ETFDH, electron transfer flavoprotein dehydrogenase. (**b**) CoQH_2_ is present in cellular membranes and lipoproteins and acts as a chain-terminating antioxidant to inhibit both the initiation and propagation steps of lipid autoxidation. CoQH_2_ or ascorbate regenerate vitamin E from the tocopheroxyl radical. (**c**) CoQ functions in the plasma membrane redox system (PMRS) with other antioxidants, such as vitamin E and ascorbate. CoQ is reduced by NAD(P)H:quinone oxidoreductase 1 (NQO1), or NADH-cytochrome *b*_5_ reductase (cyb5Red). (**d**) The CoQ reductase ferroptosis suppressor protein 1 (FSP1) regenerates CoQH_2_ at the plasma membrane to terminate lipid peroxidation and suppress ferroptosis. Created with Biorender.com (accessed on 23 May 2023).

**Figure 3 antioxidants-12-01391-f003:**
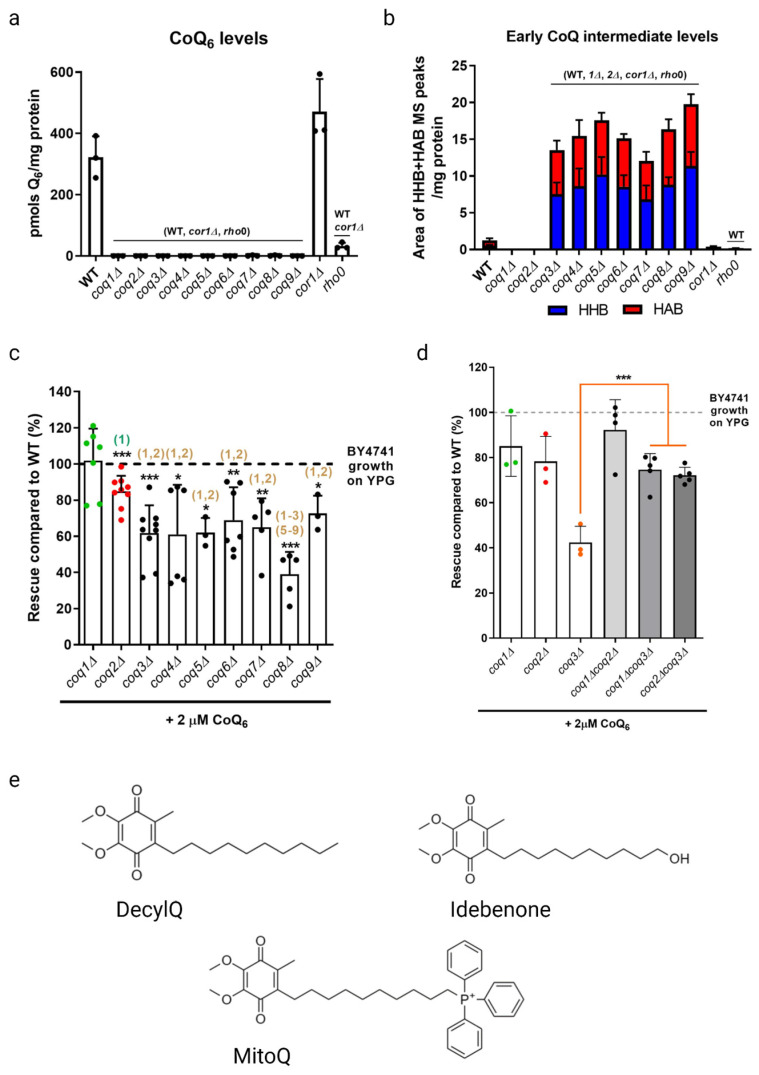
Yeast *coq* mutants are rescued with exogenous CoQ_6_ but not non-isoprenoid CoQ analogues. (**a**) CoQ_6_ was not detected in any of the *coqΔ* mutants. CoQ_6_ (pmol/mg protein) was determined based on a CoQ_6_ standard curve as described in (Tsui et al., 2019). (**b**) Each of the *coq* mutants, from *coq3Δ* to *coq9Δ*, accumulated early CoQ_6_ intermediates (HHB and HAB). The amount of HHB and HAB was calculated as the area of the MS peak/mg protein. (**c**) Yeast mutants harboring deletions in either the *COQ1* or *COQ2* gene showed more robust rescue in response to exogenous CoQ_6_ treatment than do the other single *coq* null mutants (*coq3Δ–coq9Δ*). (**d**) Additional deletions of either *COQ1* or *COQ2* restored the deficient CoQ_6_-rescue of a *coq3Δ* mutant, but do not affect the phenotype observed in *coq1Δ* or *coqΔ2* strains. Columns represent the degree of rescue (in %) ± SD of a strain compared to WT, which is defined as 100% and represented as a dashed line. Statistically significant differences between a specific *coqΔ* mutant and one of its counterparts (another *coqΔ* mutant) are denoted with numbers in parentheses on top of the columns. (1) represents differences comparing to *coq1Δ*, (2) represents differences comparing to *coq2Δ*, (3) represents differences comparing to *coq3Δ*, etc. Three or more independent rescue experiments were performed for every strain. Asterisks on top of the columns represent significant differences when compared to WT (* *p* < 0.05, ** *p* < 0.01, *** *p* < 0.001). (**e**) Structures of CoQ analogues. Panels (**a**–**d**) reproduced with permission from Fernandez-del-Rio et al. Free Radicals in Biology and Medicine, published by Elsevier, 2020 [18]. Created with Biorender.com (Accessed on 27 June 2023).

**Figure 4 antioxidants-12-01391-f004:**
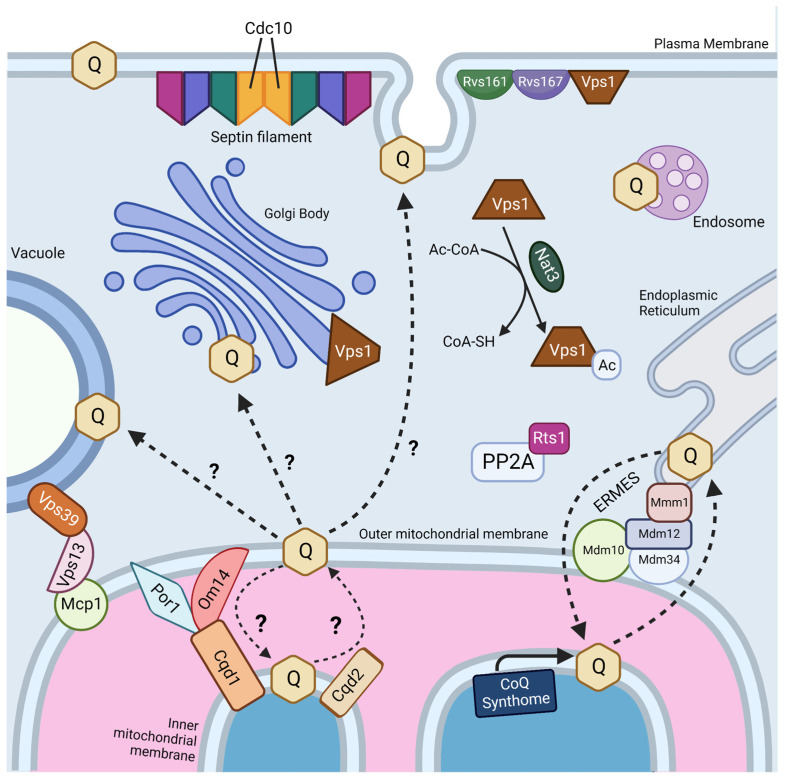
Pathways and proteins involved in trafficking of CoQ_6_ to and from mitochondria. See text for the description of gene products that are involved in transport of CoQ_6_ in yeast. The endoplasmic reticulum-mitochondria encounter structure (ERMES) is formed by Mdm10 and Mdm34 at the outer mitochondrial membrane, Mdm12 in the cytosol, and Mmm1 on the ER membrane. Septin filaments are hetero-oligomeric complexes composed of septin subunits, including Cdc10, Cdc3, Cdc12, Cdc11, and sometimes Shs1. PP2A, protein phosphatase 2A. Dashed lines denoted with question marks indicate postulated pathways of CoQ trafficking. Created with Biorender.com (Accessed on 25 May 2023).

**Figure 5 antioxidants-12-01391-f005:**
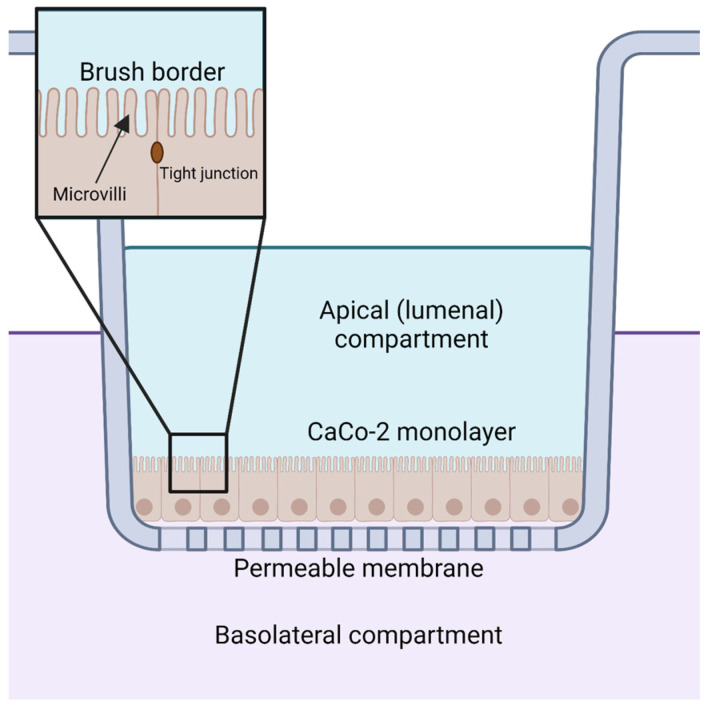
CaCo-2 cell monolayers are used to model the uptake and transport of CoQ from the apical or lumenal side to the basolateral compartment. Created with BioRender.com (Accessed on 25 May 2023).

**Figure 6 antioxidants-12-01391-f006:**
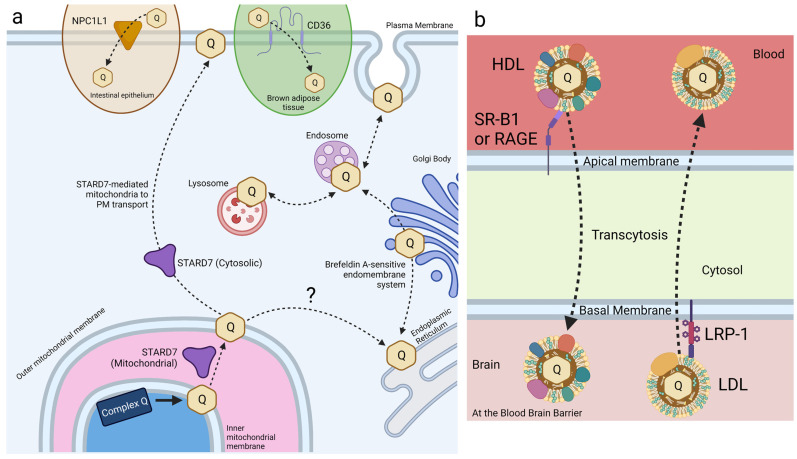
CoQ_10_ uptake and trafficking in mammalian cells. Dashed arrows indicate CoQ movement; solid arrows indicate chemical reactions. Proteins responsible for trafficking CoQ are identified where known; proteins mediating the transport shown by dashed arrows remain to be identified. (**a**) Exogenous CoQ uptake and intracellular CoQ distribution are mediated by several known mechanisms in mammalian cell culture. CD36 and NPC1L1 recognize and import exogenous CoQ within brown adipose tissue and small intestine epithelial cells, respectively. The endomembrane system facilitates intracellular trafficking of CoQ between membrane-bound vesicles. Dual-localized STARD7 in the mitochondria and cytosol support endogenous CoQ biosynthesis and distribution to the plasma membrane, respectively. (**b**) Model of lipoprotein-associated CoQ trafficking at the blood–brain barrier. SR-B1 and RAGE regulate influx of the HDL-associated CoQ across the blood–brain barrier, while LRP-1 regulates the efflux of LDL-associated CoQ. 1, receptor-mediated endocytosis; 2, recycling to apical side via the same receptor; 3, transcytosis to basolateral side; 4, transport to lysosome; 5, recycling or efflux of CoQ to apical membrane via LDL receptor family proteins; 6, paracellular transport detected due to defects in tight junctions. HDL, high-density lipoprotein; LDL, low-density lipoprotein. Created with BioRender.com (Accessed on 28 June 2023).

**Figure 7 antioxidants-12-01391-f007:**
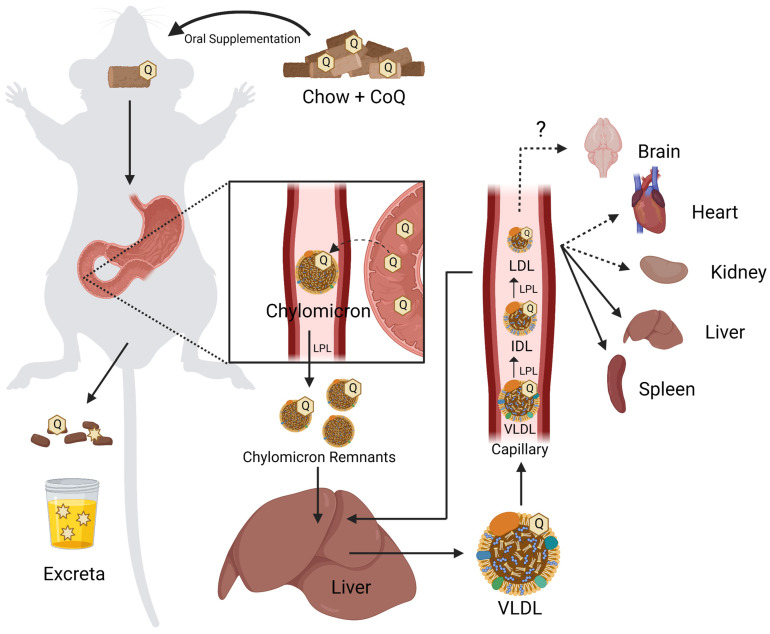
Model for the absorption of dietary-supplemented CoQ in mice. *Dashed arrows* indicate CoQ distribution to tissues enriched only at high doses or IP injection. *Stars* in urine and feces indicate metabolites of CoQ. The ? indicates an unknown mechanism for the uptake of CoQ into the brain. VLDL—very low-density lipoprotein; IDL—intermediate-density lipoprotein; LDL—low-density lipoprotein; LPL—lipoprotein lipase. Created with BioRender.com (Accessed 31 January 2023).

## Data Availability

Not applicable.

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
