# Peer review of "New Insights on the Uptake and Trafficking of Coenzyme Q"

_antioxidants, 2023, doi:10.3390/antiox12071391_

Round 1

Reviewer 1 Report

Dear authors,

Your review about the uptake and trafficking of CoQ2 could be a real masterpiece in the field of bioenergetics. I appreciate the structure of the review subdivided in specific and well-described  arguments. However, some paragraphs are heavy to read even if understandable. They should be simplified. For example the description of all yeast mutant is heavy to  read. 

English language is fine and understandable. Sometimes is more technical and heavy to read. Simplify some sentences thruogh all the paragraphs.

Author Response

Your review about the uptake and trafficking of CoQ2 could be a real masterpiece in the field of bioenergetics. I appreciate the structure of the review subdivided in specific and well-described arguments. However, some paragraphs are heavy to read even if understandable. They should be simplified. For example the description of all yeast mutant is heavy to read.

English language is fine and understandable. Sometimes is more technical and heavy to read. Simplify some sentences thruogh all the paragraphs.

We thank the Reviewer for this positive feedback about our manuscript. Sections of the manuscript that have been edited and simplified have been highlighted in yellow in the revised manuscript.

Reviewer 2 Report

The manuscript “New insights on the uptake and trafficking of Coenzyme Q” is an extensive and updated review on CoQ metabolism, uptake, and trafficking. It is well organized, and considering the subject, easy to read and follow. Only some minor changes are needed:

·       Pag2, line 81: Please consider adding choline metabolism to the list of ETFDH (alongside with dimethylglycine).

·       Pag2, line 85: there is an extra paragraph.

·       Pag5, fig2: It would be very useful to the readers to have in the figure the metabolic pathway from tyr to 4-HB. Please consider adding it.

·       Pag 7, fig3: In the legend there is no reference to d) and e). Please consider adding it.

·       Pag 9, line 342: the sentence does not continue. Something is missing and in pag10 starts with a reference. Please review it.

Author Response

The manuscript “New insights on the uptake and trafficking of Coenzyme Q” is an extensive and updated review on CoQ metabolism, uptake, and trafficking. It is well organized, and considering the subject, easy to read and follow. Only some minor changes are needed:

We thank the Reviewer for these strong and constructive comments about our manuscript.

  • Pag2, line 81: Please consider adding choline metabolism to the list of ETFDH (alongside with dimethylglycine).

We note that it would be redundant to add “choline” to line 81. This is because it is listed in line 78 as a substrate for choline dehydrogenase (CHDH). According to reference #23 and to the scheme shown in Figure 1a, CHDH oxidizes choline and transfers electrons directly to CoQ. In contrast, the components listed in line 81 use the electron transfer flavoprotein (ETF) as the electron acceptor. Thus, we prefer to restrict the mention of choline to line 78.

  • Pag2, line 85: there is an extra paragraph.

We thank the reviewer for catching this typographical error – the inappropriate paragraph return has been removed, and lines 85 and 86 have been highlighted in the revised manuscript to reflect this change.

  • Pag5, fig2: It would be very useful to the readers to have in the figure the metabolic pathway from tyr to 4-HB. Please consider adding it.

We thank the reviewer for this suggestion. Figure 2 has been revised to show the pathway from tyrosine to 4-HB. We added text that explains the new intermediates that were added to this part of the CoQ biosynthetic pathway. Please see lines 128 -142 of the revised manuscript. References #40, 41, and 42 have been added that pertain to the text explaining the synthesis of 4-HB from tyrosine.

  • Pag 7, fig3: In the legend there is no reference to d) and e). Please consider adding it.

The text for the legend to Figure 3 has been corrected, and the reference to all panels are included in the revised manuscript. In addition, Figure 3 has been slightly rearranged to enhance clarity.

  • Pag 9, line 342: the sentence does not continue. Something is missing and in pag10 starts with a reference. Please review it.

Thank you for catching this error. The missing text has been inserted and the issue is now corrected in our revised manuscript. Please see lines 350-352 in the revised manuscript.

Reviewer 3 Report

In the present review article, the authors have summarized with a critical view the current data on CoQ uptake and trafficking. The authors begin with a short introduction on the biosynthesis of CoQ and then the experimental data in various models (rodents, yeast, mammalian cell) are outlined. The manuscript contains explanatory figures that summarize the main pathways related to CoQ pathway that greatly assist the reader in following the text throughout the manuscript. Differences in the structure of CoQ (CoQ10, 9 etc) and CoQ pathways between different organisms are highlighted and the possible limitations of the reported studies in model organisms are outlined. The function of genes in biosynthesis of CoQ are analyzed in detail as well as potential defects that have been addressed through knockout technology. Interestingly such defects are causes of rare diseases. Further, complementation cloning experiment where a human gene ortholog has substituted for the yeast are outlined. These experiments have proven the importance of the CoQ pathway for the living organisms, since all the functions of the encoded proteins/enzymes are well preserved through evolution.
In addition, the authors have analyzed the data regarding the supplementation of CoQ and potential issues that relate with the bioavailability since CoQ is highly hydrophobic. Certain formulation of CoQ are presented as well as their mechanism of action.
The review is extensive, well-written and structured and I recommend the publication without any changes. There are a few typo errors e.g. line 352, lines 240-246 should be connected etc but these issues could be fixed during proofs.

Author Response

The review is extensive, well-written and structured and I recommend the publication without any changes.

We thank the Reviewer for these positive comments about our manuscript.

There are a few typo errors e.g. line 352, lines 240-246 should be connected etc but these issues could be fixed during proofs.

We thank the Reviewer for noting these errors. The missing text (formerly noted at line 352) has been replaced in the revised manuscript. Please see lines 350-352 in the revised manuscript.

 Some of the text for the Figure 3 legend was omitted and has been corrected in the revised manuscript.